# Single-cell multi-omics analysis identifies two distinct phenotypes of newly-onset microscopic polyangiitis

Masayuki Nishide [1,2,3,14] ✉, Kei Nishimura[3,4,5,14], Hiroaki Matsushita[3,4,5], Ryuya Edahiro[1,6], Sachi Inukai[5], Hiroshi Shimagami[1,2,3], Shoji Kawada[1,2,3], Yasuhiro Kato[1,2,3], Takahiro Kawasaki[1,2,3], Kohei Tsujimoto[1,2,3], Hokuto Kamon[3,4,5], Ryusuke Omiya[4,5], Yukinori Okada [6,7,8,9,10,11], Kunihiro Hattori[4,5], Masashi Narazaki[1,2,3] & Atsushi Kumanogoh [1,2,7,8,12,13] ✉

The immunological basis of the clinical heterogeneity in autoimmune vasculitis remains poorly understood. In this study, we conduct single-cell transcriptome analyses on peripheral blood mononuclear cells (PBMCs) from newly-onset patients with microscopic polyangiitis (MPA). Increased proportions of activated CD14+ monocytes and CD14+ monocytes expressing interferon signature genes (ISGs) are distinctive features of MPA. Patient-specific analysis further classifies MPA into two groups. The MPA-MONO group is characterized by a high proportion of activated CD14+ monocytes, which persist before and after immunosuppressive therapy. These patients are clinically defined by increased monocyte ratio in the total PBMC count and have a high relapse rate. The MPA-IFN group is characterized by a high proportion of ISG+ CD14+ monocytes. These patients are clinically defined by high serum interferon-alpha concentrations and show good response to immunosuppressive therapy. Our findings identify the immunological phenotypes of MPA and provide clinical insights for personalized treatment and accurate prognostic prediction.

Anti-neutrophil cytoplasmic antibody (ANCA)-associated vasculitis (AAV) is a heterogeneous autoimmune disorder characterized by the production of autoantibodies against molecules in the cytoplasm of neutrophils, such as myeloperoxidase (MPO) and proteinase 3 (PR3)[1,2]. Patients with AAV experience inflammation of small blood vessels, which leads to damage in multiple organs. Despite its diverse clinical manifestations, the immunological basis for the heterogeneity of AAV remains poorly understood. The current remission induction therapy for AAV combines glucocorticoids with immunosuppressive agents such as cyclophosphamide or rituximab[3], and recommendations for optimal treatment protocols are constantly being updated[4–7]. The presence of organ-threatening symptoms factors into drug selection, however, the treatment strategy has not been sufficiently individualized. Therefore, accurate prognostication and optimal treatment

strategies based on a comprehensive understanding of the immunological profile of individual cases could represent a paradigm shift in the management of vasculitis.

To date, different approaches such as flow cytometry, bulk transcriptomic analyses, and mouse models have been used to characterize circulating immune cells involved in AAV. The direct binding of ANCA to neutrophils is a crucial factor in the pathogenesis of AAV, as the inappropriate activation of neutrophils by ANCA results in vascular injury[8–11]. Reduced numbers and impaired function of regulatory T cells have been associated with the development of AAV[12,13]. Autoreactive CD4+ T cells that recognize MPO can induce glomerular damage in mice[14]. Removing of CD8+ T cells has been shown to reduce the severity of glomerulonephritis induced by anti-MPO antibodies in a mouse model[15]. Patients with relapsing AAV have more CD8+ effector

memory T cells, suggesting that the cytotoxic capacity of T cells plays a role in disease development and intractability[16]. Serum levels of B-cell activating factor are elevated in patients with AAV and the differentiation of B cells into antibody-producing cells is continuously promoted[17]. Dysfunction of peripheral blood cells is thus important in the pathogenesis of AAV. However, previous studies have largely been based on the classification of cell types using a limited set of cell surface markers and bulk transcriptomic profiles, which did not have sufficient sensitivity to identify cell type–specific expression differences.

Single-cell RNA sequencing (scRNA-seq) is a technique that allows for the identification of diversity within known cell populations at the single-cell level. Furthermore, multimodal single-cell approaches such as cellular indexing of transcriptomes and epitopes by sequencing (CITE-seq) have recently been developed. Integrated analysis of gene expression and surface protein markers allow us to identify and validate previously unreported subpopulations[18]. scRNA-seq has identified functional cell populations and therapeutic targets in autoimmune diseases such as rheumatoid arthritis[19], systemic lupus erythematosus[20], and systemic sclerosis[21,22]. However, single-cell-based transcriptomic analysis has not yet been reported in the context of autoimmune vasculitis.

In this work, we perform single-cell transcriptome and surface proteome analyses using CITE-seq on 109,350 peripheral blood mononuclear cells (PBMCs) and mass cytometry analysis using cytometry by time of flight (CyTOF) on 737,794 PBMCs from eight newly-onset, treatment-naïve patients with MPA and seven healthy donors. All patients underwent a physical examination and were linked with detailed clinical information including blood and urine test results and Birmingham Vasculitis Activity Score (BVAS) 2008 version 3. By visualizing the dynamics of acquired immunity underlying each case, we aim to identify multi-omics–based disease phenotypes of MPA and provide clinically applicable recommendations for predicting prognosis and selecting treatment for each patient.

## Results

### Single-cell multi-omics analysis of PBMCs derived from patients with newly-onset and treatment-naïve MPA

PBMCs were collected from eight patients with MPA and seven healthy donors. All patients had newly-onset disease, and blood samples were collected prior to the induction of immunosuppressive therapy. The clinical characteristics of each patient with MPA are shown in Supplementary Table 1. Isolated PBMCs were analyzed on a 10x chromium platform, and the transcriptome and expression of 43 surface proteins were simultaneously obtained using CITE-seq (Fig. 1a). The same samples underwent CyTOF analysis in parallel (Fig. 1a). A total of 109,350 high-quality cells were obtained for analysis. The cell populations were annotated with supervised analysis using the existing CITE-seq data[18] and divided into 28 populations. UMAP plots of PBMCs derived from healthy donors (n = 7, left) and patients with MPA (n = 8, right) are shown in Fig. 1b. UMAP plots of PBMCs obtained from a total of 15 samples (Supplementary Fig. 1a) and individual study participants (Supplementary Fig. 1b) are shown. The ratio of cell numbers in each subset to the total number of PBMCs was calculated. Among the subsets in which the average ratio was 1% or greater, increased proportions of plasmablasts and CD14+ monocytes, and decreased proportions of CD8+ naïve T cells and mucosal-associated invariant T (MAIT) cells were observed in patients with MPA (Fig. 1c). Among the other subsets, increased proportions of proliferating CD4+ T cells and decreased proportions of gamma-delta T cells (gdT), classical dendritic cells (cDC), and AXL+ dendric cells (ASDC) were observed in patients with MPA (Fig. 1d). We next conducted differential abundance analysis using Milo, a statistical framework that performs difference-in-presence tests by assigning cells to partially overlapping neighborhoods on a k-NN graph[23] (Fig. 1e). This cluster-free and age-adjusted

analysis confirmed the alterations in cellular proportions shown in Fig. 1c, d. Compared to healthy donors, the proportion of plasmablasts (median log2 fold change: +1.7), CD14+ monocytes (+0.51), and proliferating CD4+ T cells (+1.7) subsets were increased, while the proportion of CD8+ naïve T cells (median log2 fold change: −1.2), MAIT cells (−2.5), gdT cells (−0.98), cDC1 (−1.6), cDC2 (−1.6), and ASDC (−3.4) subsets were decreased in patients with MPA (Fig. 1f).

CyTOF analysis was also performed on PBMCs from the study participants. A total of 737,794 cells were analyzed and UMAP plots are shown in Supplementary Fig. 2a. Each cellular subset was annotated based on surface marker information. UMAP plots of PBMCs from each study participant are shown in Supplementary Fig. 2b. Consistent with the results of the single-cell analyses, increased proportions of CD14+ monocytes and decreased proportions of CD8+ naïve T cells were observed in patients with MPA (Supplementary Fig. 2c).

### Differential abundance analysis of monocyte, CD8+ T cell, and B cell subsets

We then focused on monocyte, CD8+ T cell, and B cell subsets and conducted differential abundance analysis using Milo. First, we subclustered monocytes into six subsets according to the RNA expression of known marker genes[24–27]: activated CD14+ monocytes (CD14 Mono_Activated), CD14+ monocytes characterized by *VCAN* gene expression (CD14 Mono_VCAN), CD14+ monocytes characterized by interferon signature gene (ISG) expression (CD14 Mono_ISG), CD14+ monocytes characterized by *HLA* gene expression (CD14 Mono_HLA), CD16+ monocytes (CD16 Mono), and classical dendritic cells (cDC) (Fig. 2a). Highly expressed genes in each subpopulation are shown in Fig. 2b. UMAP plots of monocytes derived from each study participant are shown in Supplementary Fig. 3a. Representative surface proteins in each subpopulation are shown in Supplementary Fig. 3b. Differential abundance analysis was subsequently performed to reveal the compositional changes of each neighborhood between patients with MPA and healthy donors (Fig. 2c). Compared to healthy donors, the proportion of CD14 Mono_Activated (median log2 fold change: +1.1) and CD14 Mono_ISG ( + 1.7) subsets were increased, while the proportion of CD14 Mono_HLA (median log2 fold change: −0.82), CD16 Mono (−1.2), and cDC (−1.7) subsets were decreased in patients with MPA (Fig. 2d).

The CD8+ T cell subset was similarly annotated and classified into five subpopulations according to the RNA expression of known marker genes[28,29]: naïve CD8+ T cells (CD8 T_Naïve), central memory CD8+ T cells (CD8 T_CM), effector memory CD8+ T cells (CD8 T_EM), cytotoxically active CD8+ T cells (CD8 T_CTL), and CD8+ T cells characterized by killer immunoglobulin-like receptor *(KIR)* gene expression (CD8 T_KIR) (Fig. 2e). Highly expressed genes in each subpopulation are shown in Fig. 2f. UMAP plots of CD8+ T cells from each study participant are shown in Supplementary Fig. 4a. Representative surface proteins in each population are shown in Supplementary Fig. 4b. Differential abundance analysis revealed that the proportion of CD8 T_CTL (median log2 fold changes: +0.40) and CD8 T_KIR (+0.85) subsets were increased, while the proportion CD8 T_Naïve (median log2 fold changes: −0.95) and CD8 T_EM (−0.35) subsets were decreased in patients with MPA (Fig. 2g, h). These results indicate that the characteristics of MPA include increased proportions of activated CD14+ monocytes, CD14+ monocytes with ISG expression, cytotoxically active CD8+ T cells, and KIR+ CD8+ T cells.

The B cell and antibody-producing cell subset was similarly annotated and classified into seven subpopulations according to the RNA expression of known marker genes[30,31]: naïve B cells (B_Naïve), activated naïve B cells (B_Naïve Activated), pre-switched memory B cells (B_Memory pre-switched), post-switched memory B cells (B_Memory post-switched), age-associated B cells (ABC), plasmablasts (Plasmablast), and plasma cells (Plasma cell) (Supplementary Fig. 5a). Highly expressed genes in each subpopulation are shown in Supplementary Fig. 5b. Differential abundance analysis revealed that the

proportion of B_Naïve Activated (median log$_2$ fold changes: +2.0), Plasmablast (+1.9), and Plasma cell (+1.8) subsets were increased, while the proportion of B_Memory pre-switched (median log$_2$ fold changes: −1.3) subset was decreased in patients with MPA

(Supplementary Fig. 5c, d). These results suggest that activation of B cells, characterized by an increased population of the CD69$^+$ activated naïve B cell subset and enhanced antibody production capacity, are features of MPA.

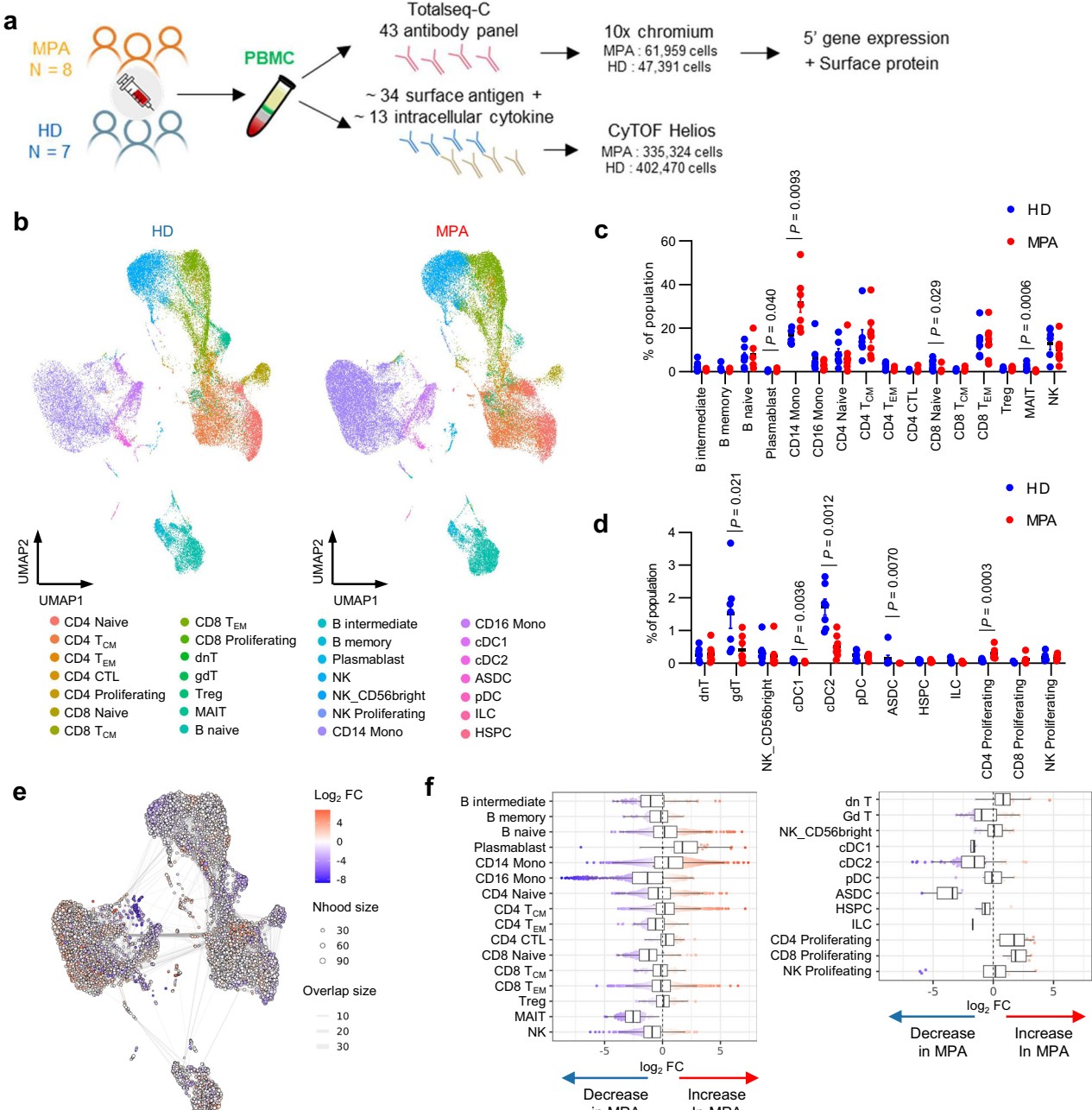

**Fig. 1 | CITE-seq analysis of PBMCs from healthy donors and patients with newly diagnosed, treatment-naïve MPA. a** Overview of the experimental workflow. MPA microscopic polyangiitis, HD healthy donors, PBMC peripheral blood mononuclear cells. **b** UMAP plots showing CITE-seq data of 47,391 PBMCs derived from healthy donors (*n* = 7, left) and 61,959 PBMCs derived from patients with MPA (*n* = 8, right). 28 cellular clusters were annotated with reference mapping. T$_{CM}$ central memory T cells, T$_{EM}$ effector memory T cells, CTL cytotoxic T lymphocytes, dnT double negative T cells, gdT gamma-delta T cells, Treg regulatory T cells, MAIT mucosal associated invariant T cells, NK natural killer cells, Mono monocytes, cDC classical dendritic cells, ASDC AXL$^+$ dendric cells, pDC plasmacytoid dendritic cells, ILC innate lymphoid cells, HSPC hematopoietic stem and progenitor cells. Percentage of each cellular subpopulation relative to total number of PBMCs derived from healthy donors (*n* = 7, blue dots) and patients with MPA (*n* = 8, red dots) for the clusters with an average ratio of 1% or greater (**c**) and less than 1% (**d**). Values are

means with SEMs and nominal *P*-values are calculated using a two-sided Mann-Whitney *U* test. **e** Neighborhood graph of monocytes using Milo differential abundance testing. Nodes represent neighborhoods from the PBMC population. Colors indicate the log$_2$-fold difference between patients with MPA and healthy donors. Neighborhoods that increased in patients with MPA are shown in red. Neighborhoods decreased in patients with MPA are shown in blue. **f** Beeswarm and box plots showing the distribution of log$_2$-fold differences in neighborhoods in different cell type clusters. Colors are represented similarly to **e**. Box plots show median and interquartile range (IQR); the lower and upper hinges correspond to the first and third quartiles. The upper whisker extends from the hinge to the largest value no further than 1.5*IQR from the hinge. The lower whisker extends from the hinge to the smallest value at most 1.5*IQR from the hinge. Source data are provided as a Source Data file.

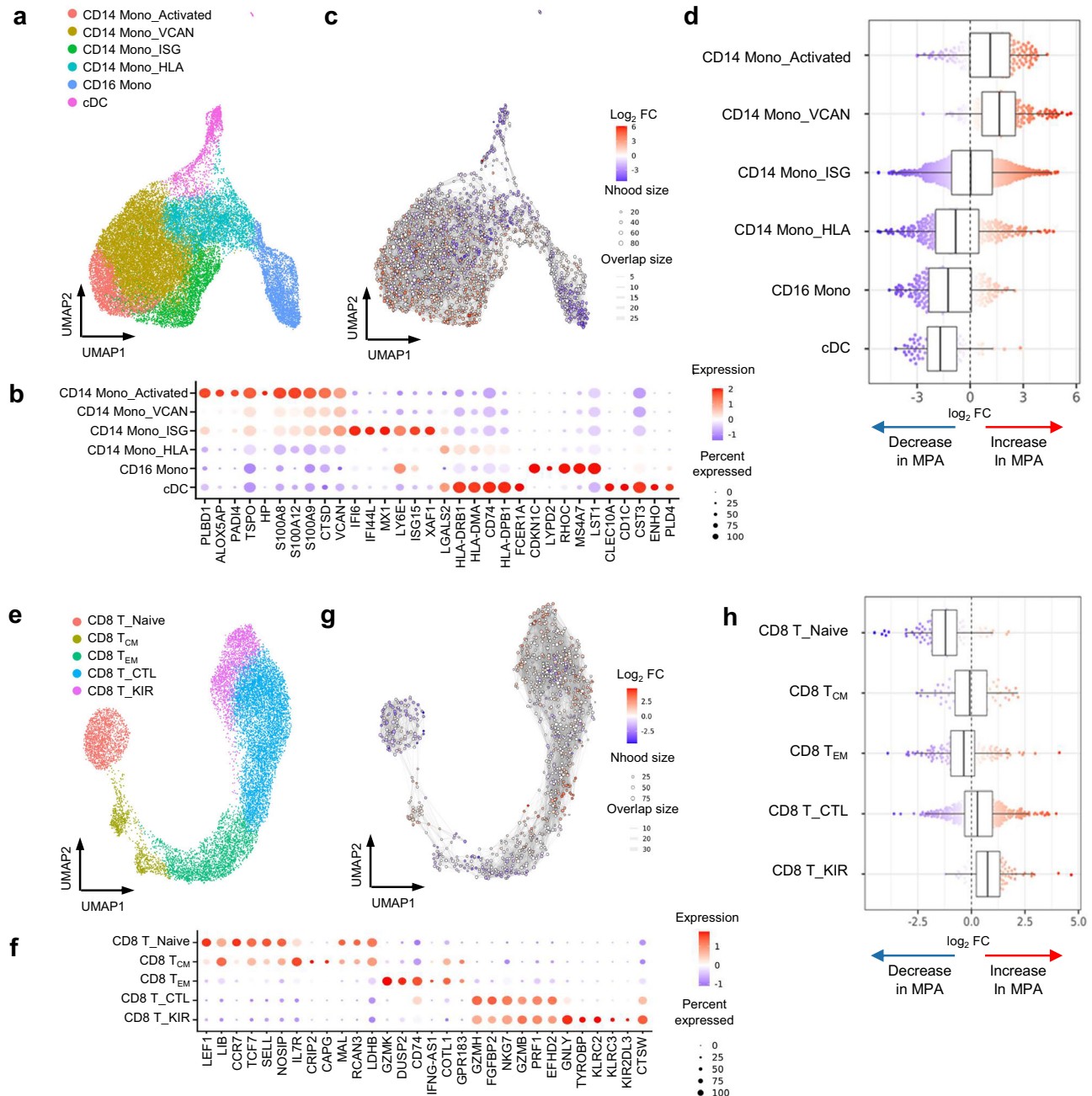

**Fig. 2 | Differential abundance analysis for the immunological characterization of MPA. a** UMAP plots showing the monocyte subpopulations in a total of 15 study participants including healthy donors (n = 7) and patients with MPA (n = 8). Six cellular clusters were identified; activated CD14⁺ monocytes (CD14 Mono_Activated), CD14⁺ monocytes characterized by *VCAN* gene expression (CD14 Mono_VCAN), CD14⁺ monocytes characterized by interferon signature gene expression (CD14 Mono_ISG), CD14⁺ monocytes characterized by *HLA* gene expression (CD14 Mono_HLA), CD16⁺ monocytes (CD16 Mono), and classical dendritic cells (cDC). **b** Balloon plot showing highly expressed genes in each monocyte subpopulation shown in **a**. Balloon color indicates the averaged scaled expression of the indicated genes. Balloon size indicates the percentage of cells expressing the indicated genes. **c** Neighborhood graph of monocytes using Milo differential abundance testing. Nodes represent neighborhoods from the monocyte population. Colors indicate the log₂-fold difference between patients with MPA and

healthy donors. Neighborhoods that increased in patients with MPA are shown in red. Neighborhoods decreased in patients with MPA are shown in blue. **d** Beeswarm and box plots showing the distribution of log₂-fold differences in neighborhoods in different cell type clusters. Colors are represented similarly to **c**. Box plots are created in a similar fashion as in Fig. 1f. **e** UMAP plots showing the CD8⁺ T cell subpopulations in 15 samples. Five cellular clusters were identified; naïve CD8⁺ T cells (CD8 T_Naïve), central memory CD8⁺ T cells (CD8 T_CM), effector memory CD8⁺ T cells (CD8 T_EM), cytotoxically active CD8⁺ T cells (CD8 T_CTL), and CD8⁺ T cells characterized by *KIR* gene expression (CD8 T_KIR). **f** Balloon plot showing highly expressed genes in each population shown in **e**. **g** Neighborhood graph of CD8⁺ T cells based on Milo differential abundance testing. The analysis was performed similarly to **c**. **h** Beeswarm and box plots of CD8⁺ T cells based on Milo differential abundance testing. The analysis was performed similarly to **d**. Box plots are created in a similar fashion as in Fig. 1f.

Differential abundance analysis was also performed on CyTOF data using the Cydar method. Based on surface marker analysis, the monocyte subsets were manually annotated and divided into four subpopulations (Supplementary Fig. 6a). In patients with MPA, the

percentage of HLA-DR⁻ CD14⁺ monocytes was higher compared to healthy donors, while the percentages of cDC, CD16⁺ monocytes, and HLA-DR⁺ CD14⁺ monocytes were lower (Supplementary Fig. 6b, c). A feature plot of representative surface proteins that identify each cell

subset is shown in Supplementary Fig. 6d. The CD8+ T cell subsets were manually annotated and divided into four subpopulations (Supplementary Fig. 7a). In patients with MPA, the percentages of CD8 T_Naïve, CM, and EM cells were lower compared to healthy donors. The percentage of CD8 T_CTL cells did not significantly change, but the cells expressing granzyme B/perforin and cells characterized by high CD57 expression (CD8 T_KIR cells) were higher in patients with MPA compared to healthy donors (Supplementary Fig. 7b–d). A feature plot of representative surface proteins or intracellular cytokines that identify each cell subset is shown in Supplementary Fig. 7d. Thus, the characteristics of the cell populations detected with CITE-seq were confirmed using molecular-based analysis.

### Distinction of transcriptome-based phenotypes using case-by-case omics analysis

To detect case-by-case differences in gene expression profiles among patients with MPA, differential expression genes (DEG) analysis was performed to identify genes with higher expression in patients with MPA than in healthy donors (Supplementary Table 2a). As a result, 40 CD14+ monocyte signature genes and 18 ISGs were included in the top-ranked DEGs (Supplementary Table 2b). Pathway analysis for DEGs revealed that CD14+ monocyte signature and interferon alpha/beta signaling pathways were enriched in patients with MPA (Fig. 3a). Subsequently, we created a heat map of study participant–specific expression of ISGs, CD14+ monocytes signature genes, and cytotoxic CD8+ T cell signature genes (Fig. 3b). Patients MPA-1 and MPA-2 had high expression of CD14+ monocytes signature genes and were therefore placed in the MPA-MONO group. Patients MPA-3, MPA-4, and MPA-5 had high expression of ISGs and were therefore placed in the MPA-IFN group. The remaining patients MPA-6, MPA-7, and MPA-8 were placed in the MPA-Others group (Fig. 3b). Gene module scores of each study participant are shown in Fig. 3c. Among participants in the MPA-IFN group, ISG expression was elevated in monocytes, CD4+ T cells, CD8+ T cells, B cells, and natural killer (NK) cells (Supplementary Fig. 8). Using the annotation and clustering shown in Fig. 2a, the ratio of cell numbers in each subset to the total number of monocytes was calculated for each case. The MPA-MONO group had a higher percentage of CD14 Mono_Activated cells (MPA-1, 23.4% and MPA-2, 17.8% of the total number of monocytes) and the MPA-IFN group had a higher percentage of CD14 Mono_ISG cells (MPA-3, 38.7%, MPA-4, 37.9%, and MPA-5, 42.6% of the total number of monocytes) than the other groups (Fig. 3d). In the CD8+ T cell and B cell subsets, there were no common changes in the cell populations that characterized each group (Fig. 3e and Supplementary Fig. 9a). Importantly, DEG analysis showed an elevated expression of MHC class II genes (e.g. HLA-DPB1, HLA-DRB1) and ISGs (e.g. IFI44L, IFITM1) in B cells from patients in the MPA-IFN group compared to those in the MPA-MONO group (Supplementary Table 3). Pathway analysis of the DEGs revealed an enrichment of MHC class II pathways and immunoglobulin production pathways in patients within the MPA-IFN group (Supplementary Fig. 9b). Thus, two phenotypes of MPA were determined based on genes expression: one characterized by CD14+ monocytes signature genes and the other characterized by enhanced ISG expression.

### Changes of transcriptome-based cell populations before and after treatment

We next compared single-cell–based cell population profiles before and after treatment in three patients; MPA-1, who was in the MPA-MONO group, MPA-3 and MPA-5, who were in the MPA-IFN group. PBMCs were collected from patient MPA-1 at the onset and at four months after the initiation of treatment, from patient MPA-3 at the onset and at twelve months after the initiation of treatment, and from patient MPA-5 at the onset and at two months after the initiation of treatment. The detailed clinical profiles of each patient are shown in Supplementary Table 4. CITE-seq data of each participant was integrated and projected in UMAP plots of monocytes (Fig. 4a). Increased population in CD14 Mono_Activated (MPA-1; before treatment, 33.5% and after treatment, 42.0%, MPA-3; before treatment, 1.42% and after treatment, 11.3%, MPA-5; before treatment, 9.53% and after treatment, 22.4% of the total number of monocytes), CD14 Mono_VCAN (MPA-1; before treatment, 46.6% and after treatment, 49.2%, MPA-3; before treatment, 21.3% and after treatment, 43.1%, MPA-5; before treatment, 25.5% and after treatment, 50.8% of the total number of monocytes), and decreased population in CD14 Mono_ISG (MPA-1; before treatment, 6.18% and after treatment, 1.77%, MPA-3; before treatment, 38.6% and after treatment, 29.3%, MPA-5; before treatment, 48.3% and after treatment, 14.4% of the total number of monocytes) were consistent across three cases, irrespective of the treatment regimen, duration, or recurrence (Fig. 4b). UMAP plots of CD8+ T cells before and after treatment were similarly generated (Fig. 4c). There were no significant changes in the CD8 T_CTL population in these two patients (MPA-1; before treatment, 35.3% and after treatment, 20.5%, MPA-3; before treatment, 16.0% and after treatment, 2.94%, MPA-5; before treatment, 60.1% and after treatment, 65.7% of the total number of CD8+ T cells; Fig. 4d).

We subsequently focused on the CD14 Mono_Activated and CD14 Mono_ISG subsets, conducting DEG analysis compared to the entire CD14+ monocyte population (Supplementary Table 5). The upregulated genes in CD14 Mono_Activated, such as FOS, ALOX5AP, and NCF1, indicate traits of immature monocytes, typically mobilized from bone marrow during inflammation[25]. Module scoring analysis confirmed the similarity of CD14 Mono_Activated to a previously reported immature monocyte subset[25] (Supplementary Fig. 10a, b). Further pathway analysis substantiated that the transcription factors CEBPD and RUNX1, known to be activated temporarily during steady-state and sepsis-induced myelopoiesis[25,32], were featured in CD14 Mono_Activated (Fig. 4e). Both DEG and pathway analyses of CD14 Mono_ISG indicated elevated levels of type I interferon-related genes, aligning with the annotated cell populations (Supplementary Table 5 and Fig. 4e). These findings suggest that the presence of CD14 Mono_Activated at the onset of MPA holds pathological significance, with an increased population of this immature monocyte subset characterizing the MPA-MONO phenotype. We also performed a comparative analysis to track genetic changes pre- and post-treatment in each patient. Genes with altered expression in CD14+ monocytes in each case are listed in Supplementary Table 6, with the $\log_2$ fold change for each gene presented in Fig. 4f. The results showed an increase in IL1R2, FKBP5, and CD163 expression in MPA-1, implying that monocyte activation and macrophage polarization[33] are characteristic during MPA-MONO relapse.

### Application to the bedside and laboratory tests

To determine whether the classification of the MPA-MONO and MPA-IFN groups could be used in real-world clinical practice, we evaluated which clinical or laboratory parameters were correlated with the CD14+ monocyte signature scores and interferon signature scores shown in Fig. 3c. The average expression of CD14+ monocyte signature genes was correlated with the percentage of monocytes among PBMCs in the complete blood count (CBC) (Supplementary Fig. 11a). The average expression of ISGs was correlated with serum IFN-α concentrations (Supplementary Fig. 11b). Therefore, we reviewed the clinical information and measured serum IFN-α concentrations in 43 patients with MPA (Supplementary Table 7). As expected, patients with MPA were divided into three groups: patients with a high monocyte ratio (MPA-MONO; green colored dots), patients with high serum IFN-α concentrations (MPA-IFN; pink colored dots), and patients with neither characteristic (Fig. 5a). Only one patient had both a high monocyte

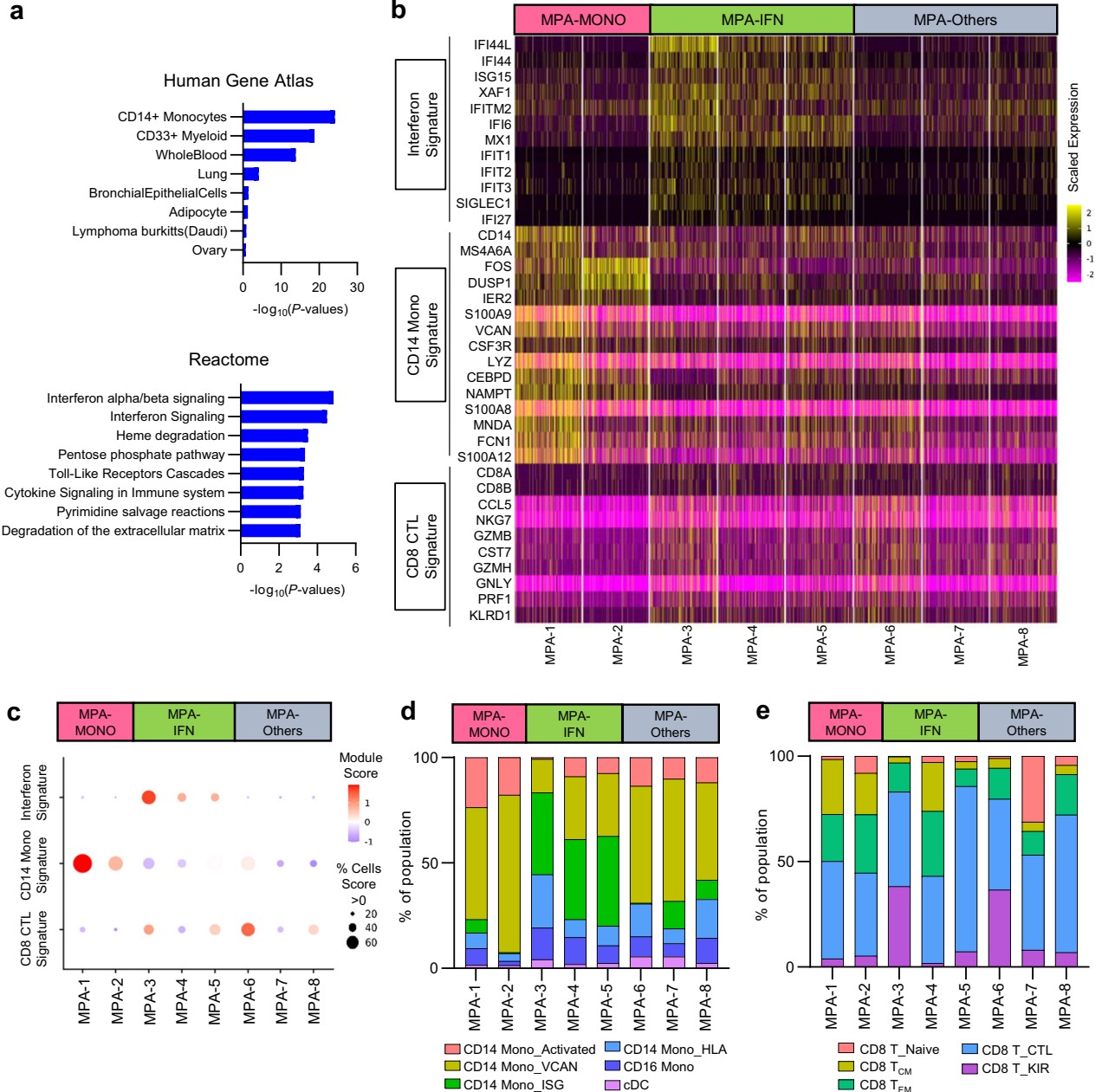

**Fig. 3 | Identification of transcriptome-based phenotypes of MPA. a** Gene set enrichment analysis of differentially expressed genes from patients with MPA using Gene Atlas from BioGPS (upper) and Reactome 2015 (bottom). *P*-values for each pathway were calculated by Benjamini−Hochberg method. **b** Heatmap showing the scaled expression of interferon signature genes, CD14$^+$ monocyte signature genes, and CD8$^+$ cytotoxic T lymphocyte (CTL) signature genes in PBMCs at the single-cell level. Each column indicates a patient with MPA (*n* = 8). **c** Balloon plots showing the averaged expression level of the signature genes shown in **b**. Each module score was calculated based on the scaled average expression level in PBMCs and the percentage of cells was calculated as the proportion of cells with a module score > 0. **d** Bar plots showing the proportion of each monocyte subpopulation relative to total the number of monocytes. Each subpopulation was annotated in a similar fashion as in Fig. 2a. **e** Bar plots showing the proportion of each CD8$^+$ T cell subpopulation relative to the total number of CD8$^+$ T cells. Each subpopulation was annotated in a similar fashion as in Fig. 2e. Source data are provided as a Source Data file.

ratio and high serum IFN-α concentrations (green and pink colored dot). In newly diagnosed patients not yet undergoing immunosuppressive therapy, MPA-MONO (3 patients) and MPA-IFN (8 patients) remained distinctly classified (Supplementary Fig. 12a). Among the BVAS components, patients in the MPA-IFN group had significantly more severe renal symptoms compared to patients in the MPA-MONO group (uncorrected *P* = 0.033) (Fig. 5b). Seven out of total nine patients in the MPA-MONO group experienced relapse, whereas only one out of nine patients in the MPA-IFN group experienced relapse. The annualized relapse rate of patients in the MPA-MONO group was significantly higher than that of patients in the MPA-IFN group (Fig. 5c).

Next, we reviewed clinical information of 43 patients with MPA to evaluate the correlation between the monocyte ratio and laboratory parameters, and between serum IFN-α concentrations and laboratory parameters (Supplementary Table 8). The monocyte ratio was positively correlated with the number of neutrophils and the levels of C-reactive proteins (Fig. 5d). Serum IFN-α levels were positively correlated with urine protein levels and serum MPO-ANCA levels (Fig. 5d). The monocyte ratio in the CBC remained almost unchanged between newly-onset cases and cases under treatment (Fig. 5e). In addition, the monocyte ratio was monitored in newly diagnosed MPA-MONO and MPA-IFN cases over a period of one year from their initial

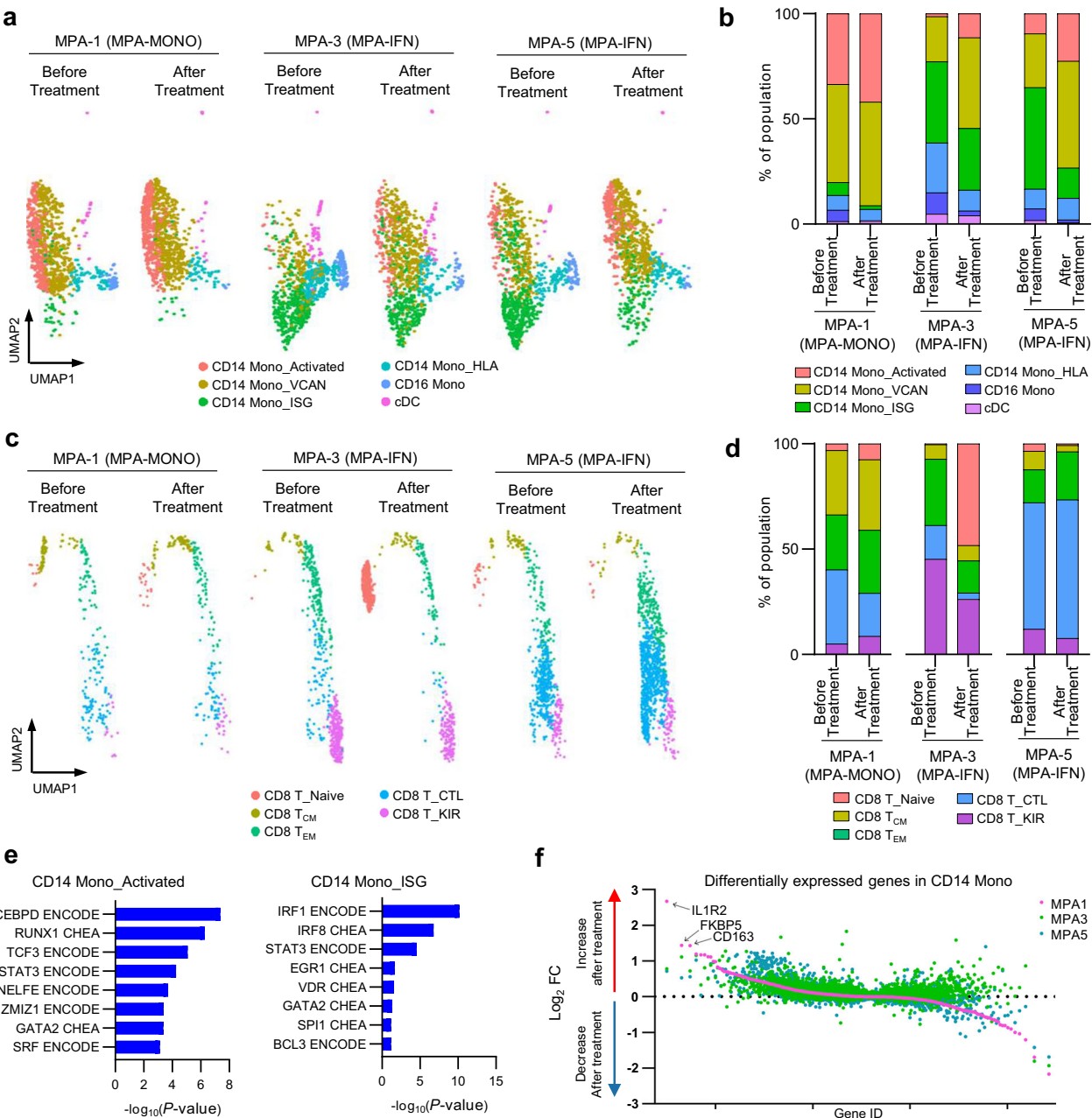

**Fig. 4 | Changes of transcriptome-based cell populations before and after treatment. a** UMAP plots showing the CITE-seq data of monocyte subpopulations before and after treatment in patients MPA-1, MPA-3, and MPA-5. Each monocyte subpopulation was annotated in a similar fashion as in Fig. 2a. **b** Bar plots showing the proportion of each monocyte subpopulation relative to the total number of monocytes before and after treatment. **c** UMAP plots showing the CITE-seq data of CD8⁺ T cell subpopulations before and after treatment in patients MPA-1, MPA-3, and MPA-5. Each CD8⁺ T cell subpopulation was annotated in a similar fashion as in Fig. 2e. **d** Bar plot showing the proportion of each CD8⁺ T cell subpopulation relative to the total number of CD8⁺ T cells. **e** Gene set enrichment analysis of differentially expressed genes in CD14 Mono_Activated (left) and CD14 Mono_ISG (right) using ENCODE and ChEA Consensus TFs from ChIP-X. *P*-values for each pathway were calculated by Benjamini–Hochberg method. **f** Scatter plot showing gene expression changes in post-treatment donors compared to pre-treatment donors. Genes are ordered by the fold change values in patient MPA-1. Pink, green, and blue colored dots represent individual genes in patients MPA-1, MPA-3, and MPA-5, respectively. Source data are provided as a Source Data file.

hospitalization (Supplementary Fig. 12b). While there was a clear difference in the monocyte ratio at baseline, no significant differences emerged over time. Serum IFN-α concentrations significantly decreased in cases under treatment (Fig. 5e). Finally, to provide prognostic insights from our cohort, we constructed a receiver operating characteristic (ROC) curve for predicting relapse from serum IFN-α concentration and percentage of monocytes in PBMC in newly diagnosed MPA patients (Fig. 5f). This ROC curve can predict the risk of

relapse before initiation of immunosuppressive treatment with a sensitivity of 82% and a specificity of 50% (Fig. 5f).

Collectively, the percentage of monocytes and serum IFN-α levels were the markers that clearly characterized the MPA-MONO and MPA-IFN groups, respectively. MPA-MONO was resistant to immunosuppressive therapy. MPA-IFN was characterized by renal symptoms and high MPO-ANCA titers and showed good response to treatment (Fig. 6).

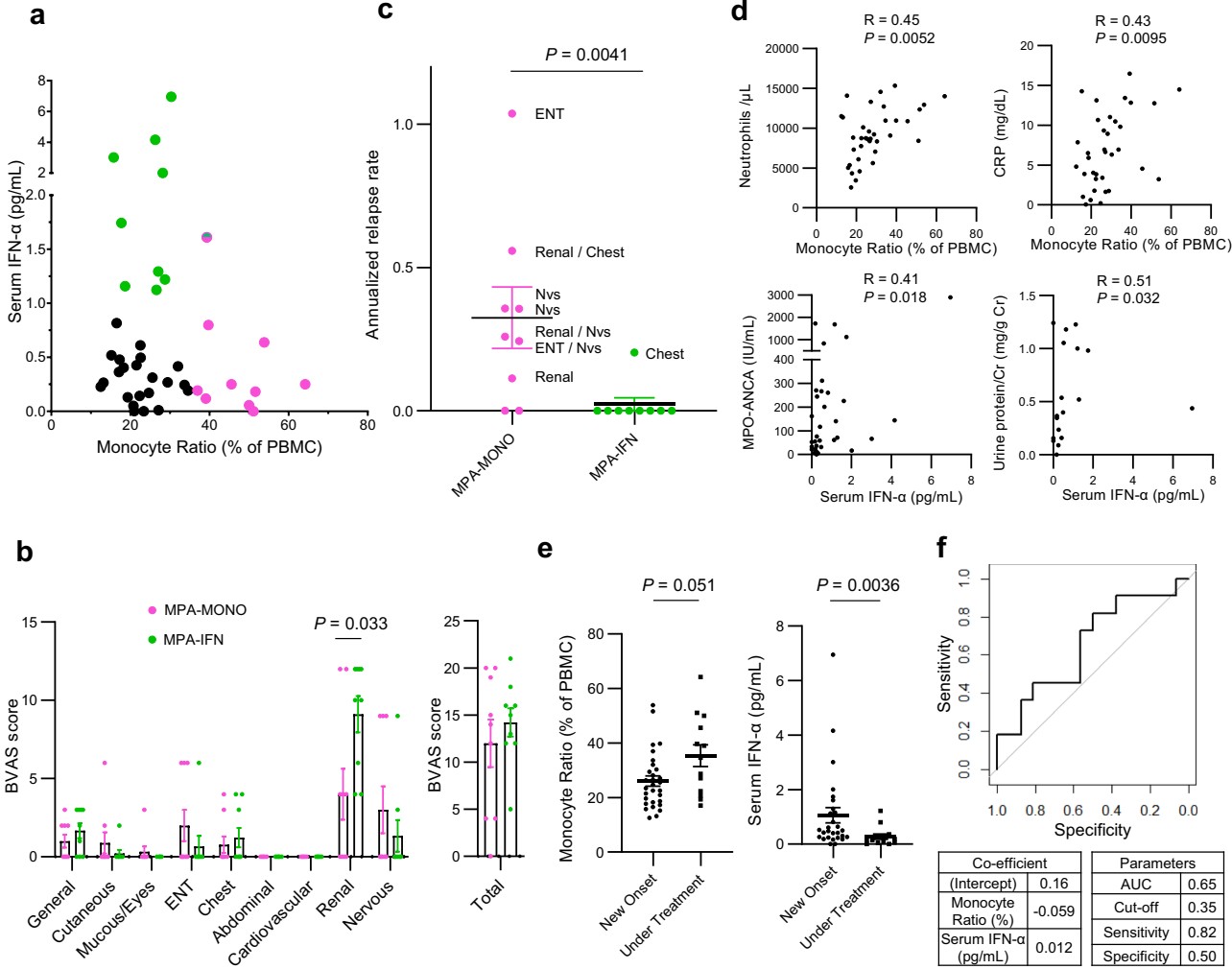

**Fig. 5 | Translating omics-based classification of MPA to the bedside. a** Serum IFN-α concentrations and the percentage of monocytes among PBMCs in the complete blood count of samples from patients with MPA (*n* = 43). The 10 samples with the highest percentage of monocytes were classified as MPA-MONO (pink colored dots). The 10 samples with the highest IFN-α concentrations were classified as MPA-IFN (green colored dots). **b** Characteristic symptoms of patients in the MPA-MONO (*n* = 9) and MPA-IFN (*n* = 9) groups. Scores for each component are based on the Birmingham Vasculitis Activity Score (BVAS) 2008 version 3. Mucous; Mucous membranes, ENT; Eyes, nose, and throat. **c** Annualized relapse rate of patients in the MPA-MONO (*n* = 9) and MPA-IFN (*n* = 9) groups. Symptoms at relapse are displayed for each case based on the components of the BVAS. "Nvs" refers to the nervous system. **d** Correlation between the percentage of monocytes and representative clinical parameters, and between serum IFN-α concentrations and representative clinical parameters. Correlations and *P*-values were quantified using Kendall's correlation coefficient (R). CRP C-reactive protein, MPO myeloperoxidase, ANCA anti-neutrophil cytoplasmic antibody. **e** Differences in serum IFN-α concentrations and monocyte ratio in patients with newly-onset cases (*n* = 30) and cases under treatment (*n* = 13). **f** Receiver Operating Characteristic (ROC) curve for predicting relapse of MPA from serum IFN-α concentration and percentage of monocytes in PBMCs. Values are means with SEMs and *P*-values are calculated using a two-sided Mann–Whitney *U* test for **b**, **c**, and **e**. Source data are provided as a Source Data file.

## Discussion

In this study, we present multi-omics analysis–based characterization of PBMC subtypes derived from patients with newly-onset, treatment-naïve MPA. Our CITE-seq results confirm the findings of previous studies, showing significant differences in the ratio of certain cell types, such as lower numbers of gamma-delta T cells, MAIT cells[34], and cDCs[35] (Fig. 1c, d). MAIT cells and gamma-delta T cells mediate early innate responses and are essential for autoimmune responses against MPO[36]. Gamma-delta T cells migrating into draining lymph nodes promote DC survival and activation. These changes in cell type ratios may be a result of the recruitment of gamma-delta T cells and DCs from peripheral vessels into tissues and lymph nodes. Among the major cell populations, we found significantly more CD14+ monocytes and fewer CD8+ naïve T cells to be characteristic of MPA (Fig. 1c). However, it was unclear how changes in the ratios of other cell groups offset these differences.

Milo, a method for differential abundance analysis, is a statistical framework that performs difference-in-presence tests by assigning cells to partially overlapping neighborhoods on a k-NN graph[23]. In this study, Milo enabled us to visualize some notable changes in gene expression–based subsets of PBMCs. Genes associated with the activation of CD14+ monocytes were highly enriched in PBMCs from patients with MPA (Fig. 2c, d). This type of monocyte exhibits a gene profile that is very similar to the profile previously reported for CD14+ monocytes that are more prevalent in patients with sepsis[25]. Furthermore, there have been reports of higher populations of CD14+ monocytes expressing Toll-like receptor (TLR) 2, TLR4, major histocompatibility complex (MHC) class II, and integrins, indicating macrophage-like activation of CD14+ monocytes in the pathogenesis of AAV[37–39]. It is widely known that ANCA production is triggered by preceding infections. Exposure to pathogens can recruit activated monocytes and be a driving force for the pathogenesis of MPA.

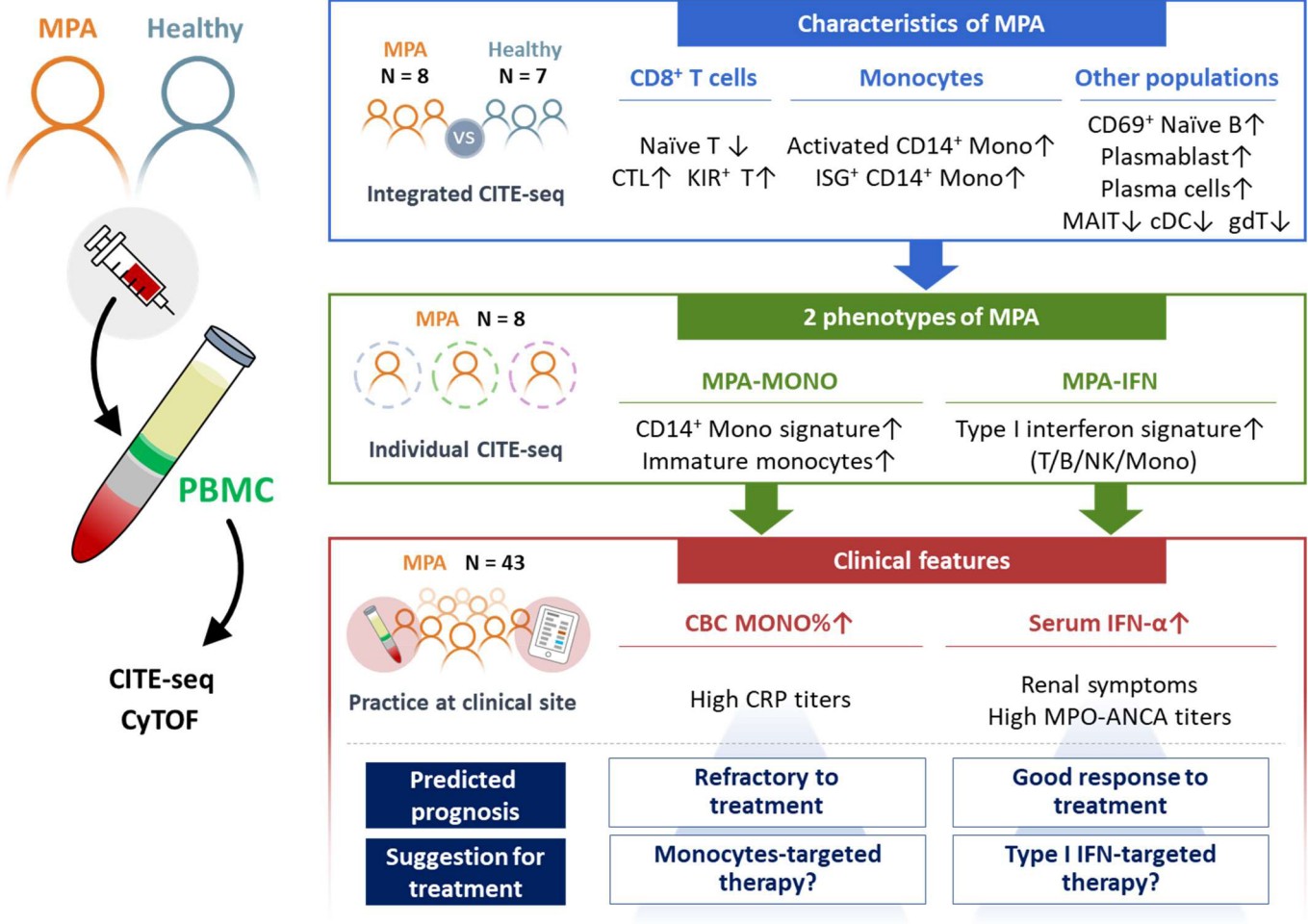

**Fig. 6 | Graphical scheme of this study.** Newly-onset, untreated patients with MPA ($n = 8$) and healthy donors ($n = 7$) were recruited for this study. MPA is characterized by increased proportions of cytotoxic CD8$^+$ T cells, KIR$^+$ CD8$^+$ T cells, activated CD14$^+$ monocytes, CD14$^+$ monocytes with ISG expression, CD69$^+$ naïve B cells, plasmablasts, and plasma cells. MPA was further subclassified into two groups based on the high expression of CD14$^+$ monocytes signature genes (MPA-MONO) or high expression of ISGs (MPA-IFN). The percentage of monocytes and serum IFN-α levels were the clinical markers that clearly distinguished MPA-MONO and MPA-IFN groups, respectively. The findings of this study suggest clinical recommendations for estimating prognosis for each patient based on the immunological phenotypes of MPA. MPA microscopic polyangiitis, PBMC peripheral blood mononuclear cells, CITE-seq cellular indexing of transcriptomes and epitopes by sequencing, CyTOF cytometry by time-of-flight, CTL cytotoxic T lymphocytes, KIR killer immunoglobulin-like receptor, ISG interferon signature genes, MAIT mucosal associated invariant T cells, cDC classical dendritic cells, gdT gamma-delta T cells, CBC complete blood count.

---

Another interesting finding is the increased population of CD14$^+$ monocytes with ISGs (Fig. 2c, d). The significance of ISGs has recently been reported in various immune diseases. In AAV, response to remission induction therapy can be predicted by monitoring the decrease in several IFN-related genes, such as *IRF7, IFIT1, IFIT5, OASL*, and *GBP-1*[40]. Importantly, we showed the ISG score is positively correlated with serum concentrations of IFN-α in patients with MPA. The primary source of interferon production in MPA is unclear, as detecting IFN-α gene expression via scRNAseq can be challenging due to the relatively low expression of these genes, as previously reported[41,42]. The aberrant activation of neutrophils in MPA may be a factor as the nucleic acid component of neutrophil extracellular traps can induce IFN activity[43].

Peripheral blood samples from patients with recurrent autoimmune diseases typically contain higher numbers of CD8$^+$ CTLs compared to healthy donors, and these cells contribute to organ damage[16]. We found that long-lived memory subsets such as T$_{CM}$; central memory T cells and T$_{EM}$; effector memory T cells were decreased in the peripheral blood of patients with MPA, while the CD8$^+$ CTL subset with high cytotoxic activity was increased (Fig. 2g, h). Interestingly, KIR$^+$ CD8$^+$ T cells, a subset of CD8$^+$ T cells that prevent immune overshoot by eliminating CD4$^+$ T cells that react abnormally to antigens[29], were significantly increased in patients with MPA. KIR$^+$ CD8$^+$ T cells play a role in suppressing autoimmune responses by recognizing and killing complexes of autopeptides bound to MHC class I antigens that are specifically present on autoreactive CD4$^+$ T cells. KIR$^+$ CD8$^+$ T cells are increased in patients with autoimmune diseases and are associated with vascular damage in patients with COVID-19[29]. Thus, KIR$^+$ CD8$^+$ T cells can be a key player in vasculitis and a promising therapeutic target for MPA.

Gene expression analysis of each patient allowed us to divide patients with MPA into two main subgroups: those with increased CD14$^+$ monocyte features and those with high ISG features (Fig. 3b, c). In this study, we referred to these phenotypes as MPA-MONO and MPA-IFN, respectively. The increase in Mono_Activated and CD14 Mono_VCAN populations, as well as the decrease in CD14 Mono_ISG population, were consistent across all three cases, regardless of the treatment regimen, duration, or recurrence (Fig. 4a, b). A previous study suggested that this type of activated CD14$^+$ monocytes may originate from bone marrow mononuclear cells rather than from mature peripheral blood cells[25]. In MPA, pathogen-induced myelopoiesis and dysregulated hematopoietic precursor differentiation may

lead to the reprogrammed monocyte population entering the bloodstream. These monocytes expressing an imprinted inflammatory condition at the bone marrow level may be responsible for resistance to oral or intravenous medication. However, post-treatment single-cell analyses were conducted on three cases with varying treatments and durations. The limited number of cases represents a study limitation due to potential immunological alterations resulting from disparate treatment regimens.

Translating of the results from large-scale single-cell analysis to clinical practice is a significant translational challenge currently. In our study, CD14$^+$ monocyte signature scores correlated with the monocyte ratio to the total number of PBMCs in a complete blood count, and ISG scores strongly correlated with serum IFN-α concentrations (Fig. 5a). Patients in the MPA-MONO group exhibited a higher rate of relapse (Fig. 5c). Patients in the MPA-IFN group had characteristics such as proteinuria, renal symptoms as assessed by the BVAS, and high titers of MPO-ANCA (Fig. 5b, d). By assessing serum IFN-α concentrations and monocyte ratios, it may be possible to predict the clinical phenotype and probability of disease recurrence in each patient. Moreover, monocyte-targeted therapies may be more suitable for the MPA-MONO phenotype, while B-cell–specific therapies or anti-IFN agents may be more appropriate for the MPA-IFN phenotype.

In summary, single-cell multi-omics analysis revealed that increased proportions of activated CD14$^+$ monocytes, CD14$^+$ monocytes characterized by ISG expression, cytotoxic CD8$^+$ T cells, and KIR$^+$ CD8$^+$ T cells were characteristic of newly diagnosed, untreated MPA. MPA was classified into two groups characterized by high expression of CD14$^+$ monocyte signature genes (MPA-MONO) and high expression of ISGs (MPA-IFN). The percentage of monocytes and serum IFN-α levels were the clinical markers that clearly characterized MPA-MONO and MPA-IFN groups, respectively. MPA-MONO is resistant to immunosuppressive therapy. MPA-IFN is characterized by renal symptoms and high MPO-ANCA titers (Fig. 6). Our findings provide insights into future therapies for vasculitis by characterizing the immunological phenotype of each patient with MPA. Further studies are needed to evaluate the roles of CD14$^+$ monocytes and CD8$^+$ T cells in the clinical course of MPA and to determine their potential as therapeutic targets.

## Methods

### Study participants
Samples were obtained after informed consent was provided by the study participant, in accordance with the Declaration of Helsinki and with approval from the ethics review board of the Graduate School of Medicine, Osaka University, Japan (No. 855). We have obtained consent to publish information including age, sex, the name of medical center, and the diagnosis. Study participants were not compensated. MPA was defined according to the 2012 Chapel Hill Consensus Conference nomenclature and definitions. Patients with AAV were diagnosed as having MPA according to the 2022 American College of Rheumatology/European Alliance of Associations for Rheumatology classification criteria[44,45]. The diagnosis was verified by at least 2 rheumatologists. The Birmingham Vasculitis Activity Score (BVAS) 2008 version 3 was used to rate MPA disease activity. We recorded the clinical status of each patient, with 31 December 2022, serving as the endpoint for this study. We calculated the individual annualized relapse rate (ARR) for each patient by dividing the number of relapses by the duration from onset to the endpoint, then converting this into an annual average.

### Patient profiles
Eight patients with MPA (four females; median age, 73 years) and seven healthy donors (four females; median age, 62 years) were recruited for CITE-seq experiments. All PBMCs samples were submitted for cytometry by time-of-flight (CyTOF) analysis as well. All patients with MPA had newly-onset disease and had not received any immunosuppressive therapy. All patients were admitted to Osaka University Hospital and underwent a comprehensive assessment to rule out potential vasculitis mimics, including infectious diseases and neoplastic lesions, before applying the 2022 ACR/EULAR MPA classification criteria. Patient MPA-1 presented with systemic symptoms, progressive interstitial pneumonia (IP), and aseptic recurrent bilateral otitis media. Patient MPA-2 presented with systemic symptoms, progressive IP, and renal dysfunction. Patients MPA-3 and MPA-8 presented with systemic symptoms and pauci-immune glomerulonephritis. Patient MPA-4 presented with systemic symptoms and extensive pachymeningitis. Patients MPA-5 and MPA-7 presented with multiple mononeuropathy. Patient MPA-6 presented with retinal vasculitis via fundoscopic examination. 43 patients with MPA (24 females; median age, 75 years) were additionally recruited to evaluate clinical and laboratory parameters. All serum samples were submitted to IFN-α ELISA.

### Serum and PBMCs preparation
Whole blood (3.5 mL) was collected in Vacutainer SST II tubes (BD Diagonostics, Cat. No. 365920). Tubes are centrifuged for 10 min at 1200×$g$. The resultant supernatant was collected as serum and stored at −80 °C. For PBMCs collection, whole blood (20 mL) was collected into a Na-heparin blood collection tube (Terumo, Cat. No. VP-H070K). PBMCs were separated using Leucosep (Greiner, Cat. No. 22788-013). PBMCs were washed and resuspended with Cellbanker 1plus (ZENOAQ, Cat. No. CB023) to a concentration of $1.0 × 10^7$ cell/mL before being stored at −150 °C.

### Single-cell library construction
PBMCs were thawed and DNA-barcoded antibodies for CITE-seq were attached. Information on the antibodies used for CITE-seq is shown in Supplementary Table 9. Single-cell suspensions were processed through the 10x Genomics Chromium Controller (10x Genomics). The libraries were constructed following the protocol outlined in the Chromium Single Cell 5′ Reagent Kits v2 (Dual Index, Cat. No. PN-1000263) User Guide (10x Genomics). Briefly, up to 10,000 labeled live cells per sample were separately loaded into the 10x Genomics platform without sample mixing to create a barcoded cDNA library for individual cells. Data quality control was performed using the Bioanalyzer (Agilent). Individual libraries were pooled for sequencing on the HiSeq 2500 or Novaseq 6000 platform (Illumina) to achieve at least 20,000 paired-end reads per cell for gene expression and 60,000 paired-end reads per cell for surface proteins. Sequence information is summarized in Supplementary Table 10.

### Reference-based and manual annotation of CITE-seq data
Raw FASTQ files were matched to the GRCh38 reference genome using CellRanger (version 6.0.6). Filtered HDF5 feature-barcode matrix files were generated using CellRanger count to establish a Seurat object. The Seurat R package (V4.2.0) was used for data quality control, scaling, transformation, clustering, dimensionality reduction, differential expression analysis, and visualization. A total of 109,350 cells were selected for further analysis out of a total of 117,791 putative cells using unique molecular identifiers (UMIs) per cells and % mitochondrial reads. Data were normalized and scaled using the SCTransform function. Cellular identity was determined by two rounds of clustering. At the first round of clustering, reference-based integration was applied for the query dataset using the CITE-seq dataset of 211,000 human PBMCs as a reference[18]. The FindTransferAnchors function was used to find anchors between the reference and the query using precomputed supervised principal component analysis (supervised PCA) transformation for SCT-normalized data. The MapQuery function was then used to transfer cell type labels and protein data from the reference to the query. Platelets and erythrocytes were removed from the analysis. To identify clusters within each major cell type, we performed a second round of clustering on monocytes (CD14 Mono, CD16 Mono, and cDC) and CD8$^+$ T cells (CD8 Naïve, CD8 T$_{CM}$, and CD8 T$_{EM}$). The

RunUMAP function was used for uniform manifold and projection (UMAP) dimensional reduction with 30 precomputed spca dimensions. A nearest-neighbor graph using the 30 dimensions of the supervided PCA reduction was computed using the FindNeighbors function followed by clustering using the FindClusters function. The newly generated UMAP was visualized using the DimPlot function. Each cluster was manually annotated using gene expression and protein data. Doublets were manually removed using cell-surface protein data (e.g., CD3, CD4, CD8, CD11c, CD19, CD56), separately.

## Differential abundance analysis using scRNA-seq data

Differential abundance analysis of patients with MPA and healthy donors was performed using scRNA-seq data. We used miloR (version 3.15) to detect sets of cells that are differentially abundant in various conditions by modeling counts of cells in the neighborhoods of a k-nearest neighbor (KNN) graph[23]. We first used the buildGraph function to construct a KNN graph based on precomputed supervised PCA with $k = 10$, using 30 principal components ($d = 30$). Next, we used the makeNhoods function to assign cells into neighborhoods based on their connectivity over the KNN graph. For computational efficiency, we subsampled 10% for monocytes and CD8$^+$ T cells. To test for differential abundance, Milo fit an NB GLM to the counts for each neighborhood, accounting for different numbers of cells across samples using TMM normalization. We included age as covariates in testNhoods function. The log$_2$ fold change of number of cells between two conditions in each neighborhood was used for visualization.

## Module scoring using scRNA-seq data

Gene scores for each study participant were visualized using the Dotplot function based on cell-based scores, which were calculated using the AddModuleScore function. Interferon signature genes (ISGs) used for module scoring were previously reported[46]. Classical monocyte signature genes and CD8$^+$ cytotoxic T lymphocytes (CTL) signature genes were determined in the human PBMC dataset[18] as genes highly expressed in the CD14$^+$ monocyte population and the CD8$^+$ CTL population, respectively. The immature monocyte signature genes and interferon-gamma signature genes were identified in accordance with previous studies[25,47].

## CyTOF assays

PBMCs were thawed and prepared to a concentration of $1 \times 10^7$ cell/mL. Next, they were cultured in RPMI-1640 medium for 6 h at 37 °C with GolgiStop supplementation (BD bioscience, Cat. No. 554724). To limit the batch effect, we barcoded each sample based on combinations of seven types of anti-CD45 antibodies obtained from MCP9 Labeling Kit (Standard Biotools, Cat. No. 201111 A) 30 min before the endpoint of the culture. Cell-ID Cisplatin (Fluidigm, Cat. No. 201064) (2 μM) was added 15 min before the endpoint of the culture. All barcoded samples were then combined and stained with antibodies specific for surface markers for 30 min at room temperature. To normalize the data across multiple batches, we combined control PBMCs (Cellular Tchnology Limited, Cat. No. CTL-UP1) across all batches. The samples were fixed with 1 mL of Maxper Fix and Perm buffer (Fluidigm, Cat. No. 201067) for 30 min at 4 °C. Cells were stained in 1 mL of Foxp3 Fixation/Permeabilization buffer (eBioscience, Cat. No. 00-5523-00) with antibodies specific for intracellular cytokines and Cell-ID intercalator-Ir (Fluidigm, Cat. No. 201192 A) for 30 min at room temperature. The antibodies used for CyTOF are shown in Supplementary Table 11. Antibodies were obtained from the Human Maxpar Direct Immune Profiling Assay (Standard Biotools, Cat. No. 201334), the Human PB Phenotyping Panel Kit (Standard Biotools, Cat. No. 201304), the Human T-Cell Phenotyping Panel Kit (Standard Biotools, Cat. No. 201305), and the Human Intracellular Cytokine I Panel Kit (Standard Biotools, Cat. No. 201308) and used at the concentrations indicated in the kits. The samples were suspended in a total of 10% Four Element Calibration Beads (Fluidigm,

Cat. No. 201078) with Cell Acquisition Solution (Fluidigm). CyTOF data were collected with a Helios CyTOF system (Fluidigm, Cat. No. 201244). Raw FCS data underwent bead-based normalization with CyTOF software (version 7.0.8493; Fluidigm).

## Normalization and population analysis of CyTOF data

In the preprocessing step, the FCS data were debarcoded by gating based on the staining patterns of anti-CD45 antibody–conjugated metals in Cytobank (https://premium.cytobank.org/cytobank/). The pre-processing gating strategy of the FCS files is shown in Supplementary Fig. 13. We used CytoNorm[48] and normalized the data across multiple batches based on a combined control sample. FlowSOM clustering was used to make 10 clusters for control samples with learning a spline to transfer from the computed 101 quantities. Combined samples were mapped with FlowSOM clustering and normalized based on the computed spline. Newly created FCS files were analyzed in Cytobank for PBMCs analysis. For each study participant, 10,532–99,958 single live cells were identified and used for further analysis. UMAP was applied to all normalized samples. Cells were manually annotated with surface proteins listed in Supplementary Table 11.

## Differential abundance analysis of CyTOF data

We used cydar (version 1.22.0)[49] to detect the set of cells that was differentially abundant in patients with MPA and healthy donors using CyTOF data. Normalized FCS files were transformed using the transformation function and used to construct hyperspheres using the countCells function (downsample = 10) with the tolerance parameter chosen so that each hypersphere had at least 50 cells, as estimated using the neighborDistances function. Hyperspheres from monocytes or CD8$^+$ T cells were then extracted and UMAP was applied to aggregated data from 15 individuals using the umap function. Enrichment of each hypersphere from patients with MPA was visualized using the ggplot function from ggplot2 (version 3.4.0).

## Pseudo-bulk differential gene expression analysis using scRNA-seq data

Differential gene expression analysis was performed between patients with MPA and healthy donors. Pseudo-bulk samples were first created by aggregating gene counts and normalized by the overall counts in individual samples. Genes whose expression rate was more than 15% in either patients with MPA or healthy donors were included in the analysis. P-values were calculated using Student's t-test. For the characterization of differential expression genes (DEG), we performed gene set enrichment analysis using Enrichr[50] for highly expressed genes in MPA (Fold Change >1.5). Human Gene Atlas from BioGPS[51] and Reactome 2015[52] were used as dataset and adjusted P-values for each pathway were calculated by Benjamini–Hochberg method. We utilized ENCODE and ChEA Consensus TFs from ChIP-X[53] to calculate the adjusted P-values for each transcription factor-related pathway of DEGs in CD14 Monocyte_Activated and CD14 Mono_ISG. DEGs in B cells were analysed using GO Biological Process 2023[54]. ISGs were identified using a gene set termed "Interferon alpha/beta signaling" and "Interferon gamma signaling" in Reactome 2015, and "Interferon Alpha Response" and "Interferon Gamma Response" from MSigDB Hallmark 2020 from GSEA[47]. CD14 Mono-signature genes were identified using a gene set termed "CD14$^+$ Monocytes" or "CD33$^+$ Myeloid" in Human Gene Atlas, and "CD14 Monocyte" and "Monocyte" in Azimuth Cell Types 2021[18].

## Measurement of serum interferon-alpha (IFN-α) levels

Serum IFN-α concentrations were measured using a pan–IFN-α ELISA detection kit (PBL Assay Science, Cat. No. 41115-1) using Flex Station3 (Molecular Devices) and analyzed using SoftMax Pro (version 7.1, Molecular Devices).

## ROC curve

The ROC curve for relapse prediction was constructed using pROC package (v1.18.0). We used the glm function and calculated the coefficient in generalized linear model (GLM) to create the combination ROC curve. The cut-off value was determined using Youden's index.

## Reporting summary

Further information on research design is available in the Nature Portfolio Reporting Summary linked to this article.

## Data availability

Count matrix data of CITE-seq are available at Genomic Expression Archive (GEA) with accession code E-GEAD-635 [https://humandbs. biosciencedbc.jp/en/hum0416-v1]. The reference for cell type annotation of PBMC in scRNA-seq was obtained from the following website (https://satijalab.org/seurat/articles/multimodal_reference_mapping. html). The GRCh38 reference genome was obtained from NCBI (https:// www.ncbi.nlm.nih.gov/assembly/GCF_000001405.26/). Source data are provided with this paper.

## Code availability

Experimental protocols and the data analysis pipeline used in our work follow the 10X Genomics and Seurat official websites. The analysis steps, functions and parameters used are described in detail in the Methods section. Custom code used in the paper is available at gihub (https://github.com/KeiNishim/MPA_scRNAseq).

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

## Acknowledgements

We would like to thank Maho Omiya, Kota Sugiyama, Riri Furuta, Takako Oya, and Kenichi Mogi for their excellent technical assistance. We also thank our colleagues at Chugai Pharmaceutical Co. Ltd., Hiroyuki Tsunoda, Kenji Yoshida, Hideaki Mizuno, Nicolas Sax, and Shoichi Met-sugi for their contributions to bioinformatics analysis. This work was financially supported by research grants from the Japan Society for the Promotion of Science (JSPS) KAKENHI (JP22K16361 to M.Ni. and JP18H05282 to A.K.), UBE foundation (to M.Ni.), Takeda Science Foun-dation (to M.Ni.), Japan Agency for Medical Research and Development (AMED) (223fa627002h0001 to A.K., JP22km0405211, JP22ek0410075, JP22km0405217, JP22ek0109594, JP223fa627002, JP223fa627010, JP233fa627011, and JP23zf0127008 to Y.O.), Japan Science and Tech-nology Agency (JST) Moonshot R&D (JPMJMS2021 and JPMJMS2024 to Y.O.), and Japan Agency for Medical Research and Development–Core Research for Evolutional Science and Technology (AMED–CREST) (22gm1810003h0001 to A.K.).

## Author contributions

M.Ni, K.N., H.M. and R.E. designed the project. M.Ni, H.S., S.K., Y.K., T.K., K.T. and M.Na recruited study participants and assessed clinical data. M.Ni, K.N., H.M. and H.S. conducted experiments. K.N. carried out data analysis. S.I. contributed to develop analysis tools. H.M., H.K., R.O., Y.O. and K.H. contributed to the preparation of materials and provided advice on project planning and data interpretation. A.K. provided funding for the study and supervised the project. M.Ni and K.N. wrote the manu-script. All authors contributed to the discussion of the results and approved the final version of the manuscript.

## Competing interests

A.K. has received grant support from Chugai Pharmaceutical Co, Ltd. K.N., H.M., S.I., H.K., R.O. and K.H. are employed by Chugai Pharma-ceutical Co, Ltd. and K.N., H.M., S.I., H.K., R.O. and K.H. also hold stocks in the company. The remaining authors declare no competing interests.

## Additional information

[1]Department of Respiratory Medicine and Clinical Immunology, Graduate School of Medicine, Osaka University, Suita, Osaka, Japan. [2]Department of Immunopathology, World Premier International Research Center Initiative (WPI), Immunology Frontier Research Center (IFReC), Osaka University, Suita, Osaka, Japan. [3]Department of Advanced Clinical and Translational Immunology, Graduate School of Medicine, Osaka University, Suita, Osaka, Japan. [4]Joint Research Chair of Innovative Drug Discovery in Immunology, World Premier International Research Center Initiative (WPI), Immunology Frontier Research Center (IFReC), Osaka University, Suita, Osaka, Japan. [5]Research Division, Chugai Pharmaceutical Co. Ltd, Yokohama, Kanagawa, Japan. [6]Department of Statistical Genetics, Graduate School of Medicine, Osaka University, Suita, Osaka, Japan. [7]Integrated Frontier Research for Medical Science Division, Institute for Open and Transdisciplinary Research Initiatives (OTRI), Osaka University, Suita, Osaka, Japan. [8]Center for Infectious Diseases for Education and Research (CiDER), Osaka University, Suita, Osaka, Japan. [9]Statistical Immunology, World Premier International Research Center Initiative (WPI), Immunology Frontier Research Center (IFReC), Osaka University, Suita, Osaka, Japan. [10]Laboratory for Systems Genetics, RIKEN Center for Integrative Medical Sciences, Yokohama, Kanagawa, Japan. [11]Department of Genome Informatics, Graduate School of Medicine, the University of Tokyo, Hongo, Bunkyo-ku, Tokyo, Japan. [12]Japan Agency for Medical Research and Development – Core Research for Evolutional Science and Technology (AMED–CREST), Osaka University, Suita, Osaka, Japan. [13]Center for Advanced Modalities and DDS (CAMaD), Osaka University, Suita, Osaka, Japan. [14]These authors contributed equally: Masayuki Nishide, Kei Nishimura. ✉e-mail: nishide@imed3.med.osaka-u.ac.jp; kumanogo@imed3.med.osaka-u.ac.jp

