## [Peer Review File · Nature Communications]

Single-cell multi-omics analysis identifies two distinct phenotypes of newly-onset microscopic polyangiitisREVIEWER COMMENTS

Reviewer #1 (Remarks to the Author):

Masayuki et al present a study on immunological subtypes of MPA patients using a comprehensive single cell multi-omics approach. They identified two major subsets of patients: one with a monocyte-signature (MPS-MONO) and one with an interferone-signature (MPA-INF). The indepth analysis results were translated into two simple parameters: the percentage of monocyte within the PBMNC and interferon-serum levels, which than were applied to a larger cohort of patients. The authors conclude that MPA-MONO were more resistant to therapy and were more prone to relapses in contrast to the MPA-INF group. They suggest that those difference might guide future therapies.

If confirmed the results of this study potentially could lead to a more personalized treatment approach in MPA.

There are some critical points which should be addressed

- there is insufficient information on patient characteristics and some of the given information is contradictory:

The authors claim to have used age matched controls. However, according to table 1 (supl) the average age of the HD is 50.6 years compared to 68.9 years in the patients. Even according to the median age stated in the text there was a difference of 10 years between patients and controls. In conclusion the HD were not properly age matched. As there are substantial age related changes in the immune system this might represent a major source of bias. Considering the huge experimental effort and that age was the only matching criterion it's curious that no properly matched controls were identified. Additional matching for sex would also have been desirable.

According to table 1 (supl) there were 4 females and 3 males. In the text (line 416) five females are mentioned.

#There are also questions concerning classification of MPA. Patient 1 and patient 4 only had ENT and constitutional symptoms. Despite the positivity for MPO-ANCA it is not likely that such patients fulfil the 2022 classification criteria for MPA. AAV patients with exclusive ENT involvement are likely to suffer from GPA rather than MPA and it is well known that

Japanese patients with GPA have a higher frequency of MPO-ANCA positivity (rather than PR3-ANCA) compared to e.g. UK-patients (Furuta et al 2017). According to CHCC patient 4 might be classified as having renal limited vasculitis rather than MPA. The authors claim to have applied the 2022 ACR /EULAR criteria. However, those criteria require testing for c-PR3-ANCA. PR3-ANCA status was not determined in 6 of 8 patients. Again, given the huge experimental effort it's hard to understand why such a simple test was not done.

#In two patients a second analysis was done during treatment. It is not clear why those two patients were chosen. Why was the analysis performed at different time points, i.e. 2 month after initiation of treatment in one patient and 4 month in the other patient?

Possibly patient 1 from the MPA-MONO group was chosen because of a relapse at that timepoint (see line 364 in the text), whereas patient MPA-5 was in remission when the second analysis was performed. Additionally, it seems very problematic, that very different therapies were used, with a classical immunosuppressant in MPA-1 (MMF) and a cytotoxic agent in MPA-5. It is likely that those different approaches will lead to different immunological changes.

#There are comparable problems with the larger MPA cohort: many patients were analyzed after initiation of medication, especially GC. GC regularly will reduce lymphocytes and therefore will induce a shift to a higher monocyte ratio. Many of the patients also have not been tested for PR3-ANCA and therefore might not be correctly classified according to the 2022 criteria.

In conclusion the authors should give a more detailed characterization of the patients, possibly including histological findings.

- Some questions concerning the interpretation of the data

#E.g. in the abstract the authors state that activated CD14+ monocytes persisted after initiation of immunosuppressive therapy. However, this was only shown in one (relapsing?) patient. In the larger cohort the monocyte / PBMNC ratio was used as a surrogate.

Correspondingly, the decrease of ISG+ CD14+ cells during treatment was shown in only one patient. It therefore can't be concluded that such cells decrease during treatment.

#The possible prognostic significance of the findings apply to only less than half of the patients as the majority of cases was in neither of the two defined groups.

Minor points

Line 85: the current use of GC and cytotoxic agents is not a result of the poor immunological understanding of AAV. Is rituximab cytotoxic?

Line 288 and fig 5b: what test was used? Was there a correction for multiple testing?

Line 291: please give details with respect to relapses and relapse rates: how long was the follow up time? What kind of relapse occurred in which group?

Line 299: according to figure 5e the monocyte ratio increased with treatment. The difference narrowly missed being statistically. I would expect a relative increase in monocytes as GC regularly reduce lymphocyte counts but not monocytes. The question therefore is in how far this is specific characteristic of the MPA-Mono group?

Reviewer #2 (Remarks to the Author):

In this study, the authors performed single-cell sequencing, surface proteome analysis, and CyTOF on peripheral blood mononuclear cells (PBMCs) from patients with microscopic polyangiitis (MPA) and healthy donors. They analyzed the resulting datasets for immune cell populations and gene expression and classified the patients into two groups. The authors also attempted to apply the classification as clinical biomarkers for efficacy prediction. While the study generated valuable datasets of patient samples and showed potential for clinical application, there are some issues that need to be addressed. The authors could have made more comprehensive use of the datasets they obtained, and further in-depth analyses are necessary to fully explore the potential of the patient-derived datasets. Moreover, the description and interpretation of the data could be improved, as some aspects of the analysis are not fully explained. Finally, the authors need to provide additional evidence to support their claim that the classification can be used as a clinical biomarker for efficacy prediction. Therefore, additional research and data analysis are necessary to fully validate the findings and to determine their clinical relevance.

Major issues:

1. The authors mainly focused on the analyses of monocytes and CD8+ T cells in patient PBMCs. Although the B cell population did not show apparent differences between MPA patients and healthy donors, it is still necessary to perform further analyses and comparison on B cell function, especially the antibody-producing ability. Importantly, the authors showed that MPO-ANCA levels were positively correlated with serum IFN- α levels, which indicated that the antibody-producing ability of B cells might differ in MPA-MONO versus MPA-IFN groups. Comparison of B cell population and gene expression between these two groups of patients are needed.

2. What is the cellular source of interferons in the setting of MPA? The authors should characterize the main cell type(s) that produce interferons by further analyses and compare the population of those interferon-producing cells in healthy donors, MPA-MONO patients and MPA-IFN patients.

3. In Fig.4, the authors only compared the cell population in patients MPA-1 and MPA-5 before and after treatment and observed a decrease in mono-ISG population. It appears that the frequency of mono-VCAN increased after treatment but the authors did not discuss this observation. In addition to the changes in monocyte and CD8+ T cell population, did the authors observe changes in gene expression in those cells? Did the patients in two different groups (MPA-MONO and MPA-IFN) show differences in gene expression and immune cell function in response to the treatment? What are the implications of these changes in biology and clinic? The authors should not simply state these changes but should interpret the data and discover the underlying biological significance and implications for clinical application. The descriptions corresponding to Fig.4 are merely a list of numbers without a clearcut and meaningful conclusion.

4. In Figure 5, why did the authors only choose IFN- α for subsequent analysis instead of the other types of interferons? Did the authors measure the concentrations of IFN- β and IFN- γ in those 43 patients? If so, do the concentrations of these two types of interferons show correlation with patient symptoms, MPO-ANCA titers and immunotherapy efficacy?

Minor issue:

The authors should provide figures with better resolution. Some genes and labels in several panels such as Fig. 3b are rather blurry to be clearly recognizable.

Our response to the Reviewer #1

We greatly appreciate the opportunity to revise our manuscript. We have responded to the comments of Reviewer #1 in the following point-by-point manner.

The line number corresponds to the number in the manuscript with tracked changes (Revised Manuscript Tracked Changed).

Masayuki et al present a study on immunological subtypes of MPA patients using a comprehensive single cell multi-omics approach. They identified two major subsets of patients: one with a monocyte-signature (MPS-MONO) and one with an interferon-signature (MPA-INF). The indepth analysis results were translated into two simple parameters: the percentage of monocyte within the PBMNC and interferon-serum levels, which than were applied to a larger cohort of patients. The authors conclude that MPA-MONO were more resistant to therapy and were more prone to relapses in contrast to the MPA-INF group. They suggest that those difference might guide future therapies.If confirmed the results of this study potentially could lead to a more personalized treatment approach in MPA.

<Our response>

We appreciate that the reviewer perfectly summarizes the essence of this study and our goals, which are "to personalize and optimize the prognostic prediction and treatment approach in MPA".

There are some critical points which should be addressed - there is insufficient information on patient characteristics and some of the given information is contradictory:

The authors claim to have used age matched controls. However, according to table 1 (supl) the average age of the HD is 50.6 years compared to 68.9 years in the patients. Even according to the median age stated in the text there was a difference of 10 years between patients and controls. In conclusion the HD were not properly age matched. As there are substantial age related changes in the immune system this might represent a major source of bias. Considering the huge experimental effort and that age was the only matching criterion it's curious that no properly matched controls were identified. Additional matching for sex would also have been desirable.

<Our response 1-1>

We appreciate the reviewer pointing out this important issue. As pointed, our study includes a broad age range, with MPA patients predominantly being elderly, while some healthy donors were relatively young. Although the difference did not have statistical significance, there are indeed discrepancies in both mean and median age. Thus, we conducted an age-adjusted analysis comparing MPA with healthy donors. Age-adjusted differential abundance (DA) analyses performed on all PBMCs (Fig. 1e, 1f) confirmed the reproducibility of changes in cell proportions observed in Fig. 1c and 1d. The DA analyses in Figures 2c, d, g, and h have already been age-adjusted. In addition, we have removed the term "age-matched" from the text and added a sentence indicating the application of age correction to the analysis.

Regarding gender matching, we apologize for the error in original manuscript. The MPA patients recruited for this cohort include four males and four females, while the healthy donors include three males and four females (Please also refer to **our response 1-2**). Although the male to female ratio isn't perfectly balanced due to the small sample size used for the single-cell analysis, we believe the sex ratio distribution is acceptable.

<Revised Points>

-Figures

We have added Fig. 1e and 1f, which include cluster-free and age-adjusted differential abundance analyses for PBMCs.

-Abstract

(Line 59 - 61)

The term 'age-matched' has been removed.

-Introduction

(Line 130 - 132)

The term 'age-matched' has also been removed.

-Results

We have added the following sentences:

(Line 163 - 172)

“We next conducted differential abundance analysis using Milo, a statistical framework that performs difference-in-presence tests by assigning cells to partially overlapping neighborhoods on a k-NN graph²² (Fig. 1e). This cluster-free and age-adjusted analysis confirmed the alterations in cellular proportions shown in Fig. 1c and 1d. Compared to healthy donors, the proportion of plasmablasts (median log₂ fold change: +1.7), CD14+ monocytes (+0.51) and proliferating CD4+ T cells (+1.7) subsets were increased, while the proportion of CD8+ naïve T cells (median log₂ fold change: -1.2), MAIT cells (-2.5), gdT cells (-0.98), cDC1 (-1.6), cDC2 (-1.6), and ASDC (-3.4) subsets were decreased in patients with MPA (Fig. 1f).”

According to table 1 (supl) there were 4 females and 3 males. In the text (line 416) five females are mentioned.

<Our response 1-2>

As mentioned in **our response 1-1**, we sincerely apologize for the error. Upon reviewing, we identified MPA-6, who is male, was inaccurately described. For the single-cell analysis, we recruited four males and four females for MPA, and for the healthy donors (HD), we recruited three males and four females. We have corrected this information in Supplementary Table 1 and in Online Methods "Patient profiles" section.

<Revised points>

-Supplementary Information

We have made revisions and corrected the gender of MPA-6 from "Female" to "Male" in Supplementary Table 1.

-Online Methods "Patient profiles"

We have revised the following sentences:

(Line 523 - 525)

"Eight patients with MPA (four females; median age, 72 years) and seven healthy donors (four females; median age, 62 years) were recruited for CITE-seq experiments."

#There are also questions concerning classification of MPA. Patient 1 and patient 4 only had ENT and constitutional symptoms. Despite the positivity for MPO-ANCA it is not likely that such patients fulfil the 2022 classification criteria for MPA. AAV patients with exclusive ENT involvement are likely to suffer from GPA rather than MPA and it is well known that Japanese patients with GPA have a higher frequency of MPO-ANCA positivity (rather than PR3-ANCA) compared to e.g. UK-patients (Furuta et al 2017). According to CHCC patient 4 might be classified as having renal limited vasculitis rather than MPA. The authors claim to have applied the 2022 ACR/EULAR criteria. However, those criteria require testing for c-PR3-ANCA. PR3-ANCA status was not determined in 6 of 8 patients. Again, given the huge experimental effort it's hard to understand why such a simple test was not done.

<Our response 1-3>

We apologize for the confusion concerning the description of PR3-ANCA titers.

Both MPO-ANCA and PR3-ANCA were measured in all cases selected for single-cell analysis. "N.D." stands for "Not Detected", indicating that the measured result was below the cutoff value. All cases tested positive for MPO-ANCA, while PR3-ANCA levels were either below or within the normal range. In response to a reviewer's comment, we have revised the PR3-ANCA value from "N.D." to "< 1.0," which is the cutoff value in our hospital (Supplementary Table 1).

We also appreciate the comments on disease classification. For all patients, we have thoroughly re-evaluated the scoring based on the 2022 ACR/EULAR criteria and provided further detail information in Supplementary Table 1 and in Online Methods "Patient profiles" section.

- All patients were admitted to Osaka University Hospital and underwent a comprehensive assessment to rule out potential vasculitis mimics.

- Before applying the 2022 ACR/EULAR MPA classification criteria, existence of the evidence for vasculitis was confirmed.

- Scoring for classification as MPA.

Furthermore, we would like to elaborate on cases MPA-1, MPA-3, and MPA-4, which the reviewer particularly noted:

MPA-1: The patient displayed general symptoms such as fever and arthralgia, progressive interstitial pneumonia (IP), and bilateral otitis media. Given the absence of nasal symptoms, positive MPO-ANCA alone, and progressive IP, we believe this case aligns with MPA according to the 2022 ACR/EULAR criteria, despite some similarities to Granulomatosis with Polyangiitis (GPA).

MPA-3: As the reviewer mentioned, the only organ-specific symptom corresponding to the BVAS score in this case is renal involvement. However, the patient's systemic symptoms, including low-grade fever, fatigue, and weight loss, which are not counted towards the BVAS score, do meet the 2022 ACR/EULAR MPA criteria.

MPA-4: This patient presented with otitis media, extensive hypertrophic pachymeningitis, positive MPO-ANCA alone, and progressive IP. Although the BVAS did not score the hypertrophic pachymeningitis due to the absence of specific neurological symptoms, we consider this case consistent with MPA.

<Revised points>

-Supplementary Information

We have added the "Classification criteria for MPA" section in Supplementary Table 1.

we have revised the PR3-ANCA value in Supplementary Table 1 from "N.D." to "< 1.0," which is the cutoff value in our hospital.

-Online Methods "Patient profiles"

We have added the following sentences:

(Line 527 - 538)

"All patients were admitted to Osaka University Hospital and underwent a comprehensive assessment to rule out potential vasculitis mimics, including infectious diseases and neoplastic lesions, before applying the 2022 ACR/EULAR MPA classification criteria. Patient MPA-1 presented with systemic symptoms, progressive interstitial pneumonia (IP), and aseptic recurrent bilateral otitis media. Patient MPA-2 presented with systemic symptoms, progressive IP, and renal dysfunction. Patients MPA-3 and MPA-8 presented with systemic symptoms and pauci-immune glomerulonephritis. Patient MPA-4 presented with systemic symptoms and extensive pachymeningitis. Patients MPA-5 and MPA-7 presented with multiple mononeuropathy. Patient MPA-6 presented with retinal vasculitis via fundoscopic examination."

#In two patients a second analysis was done during treatment. It is not clear why those two patients were chosen. Why was the analysis performed at different time points, i.e. 2 month after initiation of treatment in one patient and 4 month in the other patient? Possibly patient 1 from the MPA-MONO group was chosen because of a relapse at that timepoint (see line 364 in the text), whereas patient MPA-5 was in remission when the second analysis was performed. Additionally, it seems very problematic, that very different therapies were used, with a classical immunosuppressant in MPA-1 (MMF) and a cytotoxic agent in MPA-5. It is likely that those different approaches will lead to different immunological changes.

<Our response 1-4>

Thank you for this insightful comment. This limitation arises from the fact that blood collection for single-cell analysis was limited to hospitalization time points

(MPA-1 during relapse, and MPA-5 during routine hospitalization for IVCY). The small number of post-treatment samples and the lack of consistency in treatment type and duration represent a limitation of this study. We have addressed these limitations in the Discussion section, particularly highlighting that the variation in treatment and duration may have led to distinct immunological changes. However, we are glad if the reviewer kindly understands the scientific contribution and resource value of our data set, which is the first comprehensive single-cell data set of MPA to date including detailed clinical information.

In our earnest attempt to address this point, we obtained post-treatment data from one additional patient (MPA-3). Through this, we observed that an increase in CD14 Mono_Activated and CD14 Mono_VCAN, and a decrease in CD14 Mono_ISG, were shared characteristics in these patients, irrespective of the differences in treatment and duration.

<Revised points>

-Figures

We collected post-treatment data for an additional case (MPA-3) and conducted a time series analysis on the total of three cases (Fig. 4a, 4b, 4c, 4d).

-Results

We have added the following sentences:

(Line 303 - 314)

“Increased population in CD14 Mono_Activated (MPA-1; before treatment, 33.5% and after treatment, 42.0%, MPA-3; before treatment, 1.42% and after treatment, 11.3%, MPA-5; before treatment, 9.53% and after treatment, 22.4% of the total number of monocytes), CD14 Mono_VCAN (MPA-1; before treatment, 46.6% and after treatment, 49.2%, MPA-3; before treatment, 21.3% and after treatment, 43.1%, MPA-5; before treatment, 25.5% and after treatment, 50.8% of the total number of monocytes), and decreased population in CD14 Mono_ISG (MPA-1; before treatment, 6.18% and after treatment, 1.77%, MPA-3; before treatment, 38.6% and after treatment, 29.3%, MPA-5; before treatment, 48.3% and after treatment, 14.4% of the total number of monocytes) were consistent across three cases, irrespective of the treatment regimen, duration, or recurrence (Fig. 4b).”

-Discussion

We have added the following sentences:

(Line 473 - 476)

“However, post-treatment single-cell analyses were conducted on three cases with varying treatments and durations. The limited number of cases represents a study limitation due to potential immunological alterations resulting from disparate treatment regimens.”

#There are comparable problems with the larger MPA cohort: many patients were analyzed after initiation of medication, especially GC. GC regularly will reduce lymphocytes and therefore will induce a shift to a higher monocyte ratio. Many of the patients also have not been tested for PR3-ANCA and therefore might not be correctly classified according to the 2022 criteria.

<Our response 1-5>

The suggestion that GC treatment could influence the monocyte ratio is highly significant. As such, we created a graph for the patients who did not receive GCs, in a similar manner as in Fig. 6a (Extended Data Fig. 12a). Consequently, we confirmed that MPA-MONO (3 patients) and MPA-IFN (8 patients) were indeed distinctly classified.

As we used stored serum for the larger cohort, not many specimens had PR3-ANCA measured on the exact day of blood collection. Therefore, we reviewed the medical records of all cases and included PR3-ANCA values only in cases where PR3-ANCA was measured 'within one month prior to the date of blood collection' and 'there was no change in treatment'. This has been noted in the legend for Supplementary Table 7. Consequently, PR3-ANCA was below the cutoff in most cases. (Supplementary Table 7).

<Revised points>

-Supplementary Information

We have revised the PR3-ANCA value and the legend in Supplementary Table 7.

-Extended Data Figures

Extended Data Fig. 12a was added, which shows serum IFN- α concentrations and the percentage of monocytes among PBMCs in the complete blood count of samples from patients with MPA (newly-onset patients only, n = 28).

a

-Results

We have added the following sentences:

(Line 369 - 371)

“In newly diagnosed patients not yet undergoing immunosuppressive therapy, MPA-MONO (3 patients) and MPA-IFN (8 patients) remained distinctly classified. (Extended Data Fig. 12a).”

In conclusion the authors should give a more detailed characterization of the patients, possibly including histological findings.

<Our response>

Thank you for your constructive and clinically significant feedback. We hope our responses to each point adequately address your concerns.

- Some questions concerning the interpretation of the data

#E.g. in the abstract the authors state that activated CD14+ monocytes persisted after initiation of immunosuppressive therapy. However, this was only shown in one (relapsing?) patient. In the larger cohort the monocyte / PBMNC ratio was used as a surrogate. Correspondingly, the decrease of ISG+ CD14+ cells during treatment was shown in only one patient. It therefore can't be concluded that such cells decrease during treatment.

<Our response 1-6>

As mentioned in **Our response 1-4**, we procured time series data from an additional patient (MPA-3), and we've incorporated this patient's analysis (Fig. 4a, 4b, 4c, 4d). The results revealed an elevation in CD14 Mono_Activated, an increase in CD14 Mono_VCAN, and a decrease in CD14 Mono_ISG, which were shared features among the three cases.

<Revised points>

-Figures

We collected post-treatment data for an additional case (MPA-3) and conducted a time series analysis on the total of three cases (Fig. 4a, 4b, 4c, 4d).

-Results

We have added the following sentences:

(Line 303 - 314)

“Increased population in CD14 Mono_Activated (MPA-1; before treatment, 33.5% and after treatment, 42.0%, MPA-3; before treatment, 1.42% and after treatment, 11.3%, MPA-5; before treatment, 9.53% and after treatment, 22.4% of the total number of monocytes), CD14 Mono_VCAN (MPA-1; before treatment, 46.6% and after treatment, 49.2%, MPA-3; before treatment, 21.3% and after treatment, 43.1%, MPA-5; before treatment, 25.5% and after treatment,

50.8% of the total number of monocytes), and decreased population in CD14 Mono_ISG (MPA-1; before treatment, 6.18% and after treatment, 1.77%, MPA-3; before treatment, 38.6% and after treatment, 29.3%, MPA-5; before treatment, 48.3% and after treatment, 14.4% of the total number of monocytes) were consistent across three cases, irrespective of the treatment regimen, duration, or recurrence (Fig. 4b).”

#The possible prognostic significance of the findings apply to only less than half of the patients as the majority of cases was in nether of the two defined groups.

<Our response 1-7>

As the reviewer noted, 3 out of 8 patients in the single-cell analysis and 23 out of 43 patients in the cohort do not belong to either group. To propose findings concerning prognostic prediction based on larger cohort, we constructed a Receiver Operating Characteristic (ROC) curve to predict relapse based on monocyte cell ratio and IFN concentration. Consequently, this curve predicts the relapse risk in newly-diagnosed MPA patients with 82% sensitivity and 50% specificity (Fig. 5f)

<Revised points>

-Figures

Figure 5f, showing the ROC curve for predicting relapse from serum IFN- α concentration and percentage of monocytes in PBMC in newly diagnosed MPA patients, was added.

-Results

We have added the following sentences:

(Line 390 - 395)

“Finally, to provide prognostic insights from our cohort, we constructed a receiver operating characteristic (ROC) curve for predicting relapse from serum IFN- α concentration and percentage of monocytes in PBMC in newly diagnosed MPA patients (Fig. 5f). This ROC curve can predict the risk of relapse before initiation of immunosuppressive treatment with a sensitivity of 82% and a specificity of 50% (Fig. 5f)”

-Online Methods “ROC curve”

We have added the following sentences:

(Line 690 - 694)

“The ROC curve for relapse prediction was constructed using pROC package (v1.18.0). We used the glm function and calculated the coefficient in generalized linear model (GLM) to create the combination ROC curve. The cut-off value was determined using Youden's index.”

Minor points

Line 85: the current use of GC and cytotoxic agents is not a result of the poor immunological understanding of AAV. Is rituximab cytotoxic?

<Our response 1-8>

We have revised the statement accordingly. Several references have also been included for essential evidence supporting the treatment protocols.

<Revised points>

-Introduction

We have revised the following sentences:

(Line 88 - 95)

“The current remission induction therapy for AAV combines glucocorticoids with immunosuppressive agents such as cyclophosphamide or rituximab³, and recommendations for optimal treatment protocols are constantly being updated⁴⁻⁷. The presence of organ-threatening symptoms factors into drug selection, however, the treatment strategy has not been sufficiently individualized.”

-References

We have added the following references:

3 Hellmich, B. et al. EULAR recommendations for the management of ANCA-associated vasculitis: 2022 update. Ann Rheum Dis, doi:10.1136/ard-2022-223764 (2023).

4 de Groot, K. et al. Pulse versus daily oral cyclophosphamide for induction of remission in antineutrophil cytoplasmic antibody-associated vasculitis: a randomized trial. Ann Intern Med 150, 670-680, doi:10.7326/0003-4819-150-10-200905190-00004 (2009).

5 Jones, R. B. et al. Rituximab versus cyclophosphamide in ANCA-

associated renal vasculitis. N Engl J Med 363, 211-220, doi:10.1056/NEJMoa0909169 (2010).

6 Stone, J. H. et al. Rituximab versus cyclophosphamide for ANCA-associated vasculitis. N Engl J Med 363, 221-232, doi:10.1056/NEJMoa0909905 (2010).

7 Furuta, S. et al. Effect of Reduced-Dose vs High-Dose Glucocorticoids Added to Rituximab on Remission Induction in ANCA-Associated Vasculitis: A Randomized Clinical Trial. JAMA 325, 2178-2187, doi:10.1001/jama.2021.6615 (2021).

Line 288 and fig 5b: what test was used? Was there a correction for multiple testing?

<Our response 1-9>

The Mann-Whitney U test was applied to each BVAS component, and we did not correct for multiple testing for this evaluation. We have clarified that uncorrected p values were presented in the figure.

<Revised points>

-Figures

Uncorrected p value is presented in Fig. 5b.

-Results

We have revised the following sentences:

(Line 371 - 373)

Among the BVAS components, patients in the MPA-IFN group had significantly more severe renal symptoms compared to patients in the MPA-MONO group (uncorrected $p = 0.033$) (Fig. 5b).

Line 291: please give details with respect to relapses and relapse rates: how long was the follow up time? What kind of relapse occurred in which group?

<Our response 1-10>

We apologize for the lack of clarity in describing the Annual Relapse Rate (ARR) calculation. Additional information has been added into the Methods section. For the ARR, we considered the number of relapses for each patient until December

31, 2022, as the observation end date. We divided the number of relapses by the follow-up period, converting the relapse rate for each patient into 365-day units, and presented in each graph as individual ARR. Furthermore, we have included the symptoms at relapse for each group in Fig. 5c. Of note, 4 out of 7 MPA-MONO patients experienced relapse with nervous symptoms. We appreciate the reviewer for highlighting this important aspect.

<Revised points>

-Figures

Symptoms at relapse are displayed in Fig. 5c, for each case based on the components of the BVAS.

-Online Methods “Study participants”

We have revised the following sentences:

(Line 516 - 520)

“We recorded the clinical status of each patient, with December 31, 2022, serving as the endpoint for this study. We calculated the individual annualized relapse rate (ARR) for each patient by dividing the number of relapses by the duration from onset to the endpoint, then converting this into an annual average.”

Line 299: according to figure 5e the monocyte ratio increased with treatment. The difference narrowly missed being statistically. I would expect a relative increase in monocytes as GC regularly reduce lymphocyte counts but not monocytes. The question therefore is in how far this is specific characteristic of the MPA-Mono group?

<Our response 1-11>

We analyzed the monocyte ratio in MPA-MONO (3 patients) and MPA-IFN (8 patients), in the group of patients not treated with GC, tracked over a one-year period since their first hospitalization (Extended Data Fig. 12b). While there was a clear difference in the monocyte ratio at baseline, no significant differences emerged as the observation proceeded. As the reviewer suggested, we thoroughly discussed whether the observed increase in the monocyte ratio in treated cases is specific to MPA type (Fig. 5e). However, we do not consider it a defining characteristic of either the MPA-MONO or MPA-IFN types.

<Revised points>

-Extended Data Figures

Extended Data Fig. 12b was added, which shows the monocyte ratio in newly-diagnosed MPA-MONO (n = 3) and MPA-IFN (n = 8) patients, tracked over a one-year period since first hospitalization.

b

-Results

We have added the following sentences:

(Line 385 - 389)

“In addition, the monocyte ratio was monitored in newly diagnosed MPA-MONO and MPA-IFN cases over a period of one year from their initial hospitalization (Extended Data Fig. 12b). While there was a clear difference in the monocyte ratio at baseline, no significant differences emerged over time.”

Our response to the Reviewer #2

We greatly appreciate the opportunity to revise our manuscript. We have responded to the comments of Reviewer #2 in the following point-by-point manner.

The line number corresponds to the number in the manuscript with tracked changes (Revised Manuscript Tracked Changed).

In this study, the authors performed single-cell sequencing, surface proteome analysis, and CyTOF on peripheral blood mononuclear cells (PBMCs) from patients with microscopic polyangiitis (MPA) and healthy donors. They analyzed the resulting datasets for immune cell populations and gene expression and classified the patients into two groups. The authors also attempted to apply the classification as clinical biomarkers for efficacy prediction. While the study generated valuable datasets of patient samples and showed potential for clinical application, there are some issues that need to be addressed. The authors could have made more comprehensive use of the datasets they obtained, and further in-depth analyses are necessary to fully explore the potential of the patient-derived datasets. Moreover, the description and interpretation of the data could be improved, as some aspects of the analysis are not fully explained. Finally, the authors need to provide additional evidence to support their claim that the classification can be used as a clinical biomarker for efficacy prediction. Therefore, additional research and data analysis are necessary to fully validate the findings and to determine their clinical relevance.

<Our response>

We are grateful to the reviewer for acknowledging the value of the data set obtained in this study and its potential for future clinical applications. However, the reviewer also highlighted the need for a more detailed interpretation and explanation of the data set. We appreciate the scientific comments from this reviewer, and address the concerns raised in the following point-by-point responses.

Major issues:

1. The authors mainly focused on the analyses of monocytes and CD8+ T cells in patient PBMCs. Although the B cell population did not show apparent

differences between MPA patients and healthy donors, it is still necessary to perform further analyses and comparison on B cell function, especially the antibody-producing ability. Importantly, the authors showed that MPO-ANCA levels were positively correlated with serum IFN- α levels, which indicated that the antibody-producing ability of B cells might differ in MPA-MONO versus MPA-IFN groups. Comparison of B cell population and gene expression between these two groups of patients are needed.

<Our response 2-1>

We appreciate the reviewer's suggestion to further analyze the gene expression and functionality changes in B cells and antibody-producing cells, particularly in MPA-IFN cases.

In response to this constructive advice, we carried out additional mapping and annotation of B cells and antibody-producing cells, as well as a differential abundance (DA) analysis (Extended Data Fig. 5). When refining the analysis for the B-cell population, additional manual removal of low-quality cells with a low number of detected genes was performed and presented as revised UMAP (Fig. 1b). Consequently, we observed an increase in the plasmablasts population in reference mapping (Fig 1c, 1d;). CD69⁺ activated naïve B cells, plasmablasts, and plasma cells are characteristic of MPA in DA analysis (Extended Data Fig. 5).

Furthermore, we evaluated the proportion of each B cell subset across the eight MPA cases. Although there were no consistent shifts in the proportions of each B cell subset (Extended Data Fig. 9a), DEG analysis showed an elevated expression of MHC class II genes (e.g. *HLA-DPB1*, *HLA-DRB1*) and ISGs (e.g. *IFI44L*, *IFITM1*) in B cells from patients in the MPA-IFN group compared to those in the MPA-MONO group (Supplementary Table 3). Pathway analysis of these DEGs further revealed an enrichment of MHC class II and immunoglobulin production pathways in patients within the MPA-IFN group (Extended Data Fig. 9b), suggesting an increased activity of autoantibody production.

<Revised points>

- Figures

Mapping of Fig. 1b and quantitative data of Fig. 1c was revised, to refine UMAP

plots of B cells and antibody-producing cells.

Information on B cells were added in Fig. 6, a graphic summary for this work.

-Extended Data Figures

Extended Data Fig. 5 was added, which shows UMAP plots and differential

abundance analysis of B cells.

Extended Data Fig. 9a was added, which shows the population analysis of B cells derived from each study participant.

Extended Data Fig. 9b was added, which shows the pathway analysis of the DEGs in patients within the MPA-IFN group.

b

-Supplementary Information

Supplementary Table 3 was added, which shows DEG analysis in B cells from patients in the MPA-IFN group compared to those in the MPA-MONO group.

-Results

We have revised the following sentences:

(Line 157 - 160)

“Among the subsets in which the average ratio was 1% or greater, increased proportions of plasmablasts and CD14⁺ monocytes, and decreased proportions of CD8⁺ naïve T cells and mucosal associated invariant T (MAIT) cells were observed in patients with MPA (Fig. 1c).”

We have added the following sentences:

(Line 220 - 234)

“The B cell and antibody-producing cell subset was similarly annotated and classified into seven subpopulations according to the RNA expression of known marker genes^{29,30}: naïve B cells (B_Naïve), activated naïve B cells (B_Naïve Activated), pre-switched memory B cells (B_Memory pre-switched), post-switched memory B cells (B_Memory post-switched), age-associated B cells (ABC), plasmablasts (Plasmablast), and plasma cells (Plasma cell) (Extended Data Fig. 5a). Highly expressed genes in each subpopulation are shown in Extended Data Fig. 5b. Differential abundance analysis revealed that the

proportion of B_Naïve Activated (median log² fold changes: +2.0), Plasmablast (+1.9), and Plasma cell (+1.8) subsets were increased, while the proportion of B_Memory pre-switched (median log² fold changes: -1.3) subset was decreased in patients with MPA (Extended Data Fig. 5c, 5d). These results suggest that activation of B cells, characterized by an increased population of the CD69+ activated naïve B cell subset and enhanced antibody production capacity, are features of MPA”

We have further added the following sentences:

(Line 279 - 287)

“In the CD8⁺ T cell and B cell subsets, there were no common changes in the cell populations that characterized each group (Fig. 3e, Extended Data Fig. 9a). Importantly, DEG analysis showed an elevated expression of MHC class II genes (*HLA-DPB1*, *HLA-DRB1*) and ISGs (*IFI44L*, *IFITM1*) in B cells from patients in the MPA-IFN group compared to those in the MPA-MONO group (Supplementary Table 3). Pathway analysis of the DEGs revealed an enrichment of MHC class II pathways and immunoglobulin production pathways in patients within the MPA-IFN group (Extended Data Fig. 9b).”

2. What is the cellular source of interferons in the setting of MPA? The authors should characterize the main cell type(s) that produce interferons by further analyses and compare the population of those interferon-producing cells in healthy donors, MPA-MONO patients and MPA-IFN patients.

<Our response 2-2>

We share the reviewer's interest in the cellular source of IFN. However, as previously reported^{40,41}, it's challenging to target IFN genes in scRNAseq due to their relatively low expression. While the 10x platform is the most popular platform due to its high throughput quality, it also has the aspect of sparsity of data. Despite our best efforts, we found it difficult to identify interferon-producing cells in our dataset. This point has been added to the Discussion section with reference to the previous studies.

<Revised points>

-Discussion

We have added the following sentences:

(Line 435 - 439)

“The primary source of interferon production in MPA is unclear, as detecting IFN- α gene expression via scRNAseq can be challenging due to the relatively low expression of these genes, as previously reported^{40,41}.”

-References

We have added the following references:

40 Wimmers, F. et al. Single-cell analysis reveals that stochasticity and paracrine signaling control interferon-alpha production by plasmacytoid dendritic cells. Nat Commun 9, 3317, doi:10.1038/s41467-018-05784-3 (2018).

41 Bibby, J. A. et al. Systematic single-cell pathway analysis to characterize early T cell activation. Cell Rep 41, 111697, doi:10.1016/j.celrep.2022.111697 (2022).

3. In Fig.4, the authors only compared the cell population in patients MPA-1 and MPA-5 before and after treatment and observed a decrease in mono-ISG population. It appears that the frequency of mono-VCAN increased after treatment but the authors did not discuss this observation. In addition to the changes in monocyte and CD8+ T cell population, did the authors observe changes in gene expression in those cells? Did the patients in two different groups (MPA-MONO and MPA-IFN) show differences in gene expression and immune cell function in response to the treatment? What are the implications of these changes in biology and clinic? The authors should not simply state these changes but should interpret the data and discover the underlying biological significance and implications for clinical application. The descriptions corresponding to Fig.4 are merely a list of numbers without a clearcut and meaningful conclusion.

<Our response 2-3>

The reviewer suggested that our pre- and post-treatment time series analysis should track more detailed gene expression changes within each cell population, rather than merely noting shifts in the proportion of these populations.

In response to this valuable suggestion, we have made the following improvements:

i) We collected post-treatment data for an additional case (MPA-3) and conducted a time series analysis on a total of three cases (Fig. 4a, 4b, 4c, 4d). This analysis revealed shared characteristics, such as an increase in CD14 Mono_Activated and CD14 Mono_VCAN, and a decrease in CD14 Mono_ISG, irrespective of the treatment regimen, duration, or recurrence.

ii) Given that CD14 Mono_VCAN is a population possessing a genetic profile intermediate between CD14 Mono_Activated and CD14 Mono_ISG (Fig. 2b), its proportion changes in response to increases or decreases in CD14 Mono_Activated and CD14 Mono_ISG populations. Therefore, we performed more detailed differential gene expression (DEG) and pathway analysis on the CD14 Mono_Activated and CD14 Mono_ISG cell populations (Supplementary Table 5, Fig. 4e). Based on the distinctive gene expression profile, CD14 Mono_Activated appears to represent an inflammatory monocyte population, newly recruited from the bone marrow in response to inflammatory conditions. These findings suggest that the presence of CD14 Mono_Activated at the onset of MPA holds pathological significance, and an increased population of this immature monocyte subset characterizes the MPA-MONO phenotype. We believe this approach provided more insight than merely listing proportions of cell populations.

iii) We further analyzed the variations in gene expression of entire CD14⁺ monocyte populations before and after treatment. This led to a discussion on the potential contribution of *IL1R2*, *FKBP5*, and *CD163* expression in MPA-1, suggesting that monocyte activation and macrophage polarization are characteristic during MPA-MONO relapse (Fig. 4f).

<Revised points>

-Figures

We collected post-treatment data for an additional case (MPA-3) and conducted a time series analysis on the total of three cases (Fig. 4a, 4b, 4c, 4d).

Fig. 4e, which shows gene set enrichment analysis of differentially expressed genes in CD14 Mono_Activated and CD14 Mono_ISG, was added.

Fig. 4f, which shows gene expression changes between pre- and post-treatment donors, was added.

-Supplementary Information

Supplementary Table 5, which shows the list of highly expressed genes in CD14 Mono_Activated and CD14 Mono_ISG subsets, was added.

Supplementary Table 6, which shows the list of genes with altered expression before and after treatment for each case, was added.

-Results

We have added the following sentences:

(Line 303 - 314)

“Increased population in CD14 Mono_Activated (MPA-1; before treatment, 33.5% and after treatment, 42.0%, MPA-3; before treatment, 1.42% and after treatment, 11.3%, MPA-5; before treatment, 9.53% and after treatment, 22.4% of the total number of monocytes), CD14 Mono_VCAN (MPA-1; before treatment, 46.6% and after treatment, 49.2%, MPA-3; before treatment, 21.3% and after treatment, 43.1%, MPA-5; before treatment, 25.5% and after treatment, 50.8% of the total number of monocytes), and decreased population in CD14 Mono_ISG (MPA-1; before treatment, 6.18% and after treatment, 1.77%, MPA-3; before treatment, 38.6% and after treatment, 29.3%, MPA-5; before treatment, 48.3% and after treatment, 14.4% of the total number of monocytes) were consistent across three cases, irrespective of the treatment regimen, duration, or recurrence (Fig. 4b).”

We have further added the following sentences:

(Line 326 - 347)

“We subsequently focused on the CD14 Mono_Activated and CD14 Mono_ISG subsets, conducting DEG analysis compared to the entire CD14⁺ monocyte population (Supplementary Table 5). The upregulated genes in CD14 Mono_Activated, such as *FOS*, *ALOX5AP*, and *NCF1*, indicate traits of immature monocytes, typically mobilized from bone marrow during inflammation²⁴. Module scoring analysis confirmed the similarity of CD14 Mono_Activated to a previously reported immature monocyte subset²⁴ (Extended Data Fig. 10a, 10b). Further pathway analysis substantiated that the transcription factors *CEBPD* and *RUNX1*, known to be activated temporarily during steady-state and sepsis-induced myelopoiesis^{24,31}, were featured in CD14 Mono_Activated (Fig. 4e). Both DEG and pathway analyses of CD14 Mono_ISG indicated elevated levels of type I interferon-related genes, aligning

with the annotated cell populations (Supplementary Table 5, Fig. 4e). These findings suggest that the presence of CD14 Mono_Activated at the onset of MPA holds pathological significance, with an increased population of this immature monocyte subset characterizing the MPA-MONO phenotype. We also performed a comparative analysis to track genetic changes pre- and post-treatment in each patient. Genes with altered expression in CD14⁺ monocytes in each case are listed in Supplementary Table 6, with the log₂ fold change for each gene presented in Fig. 4f. The results showed an increase in *IL1R2*, *FKBP5*, and *CD163* expression in MPA-1, implying that monocyte activation and macrophage polarization³² are characteristic during MPA-MONO relapse.”

-Online Methods “Module scoring using scRNA-seq data”

We have further added the following sentences:

(Line 617 - 619)

“The immature monocyte signature genes and interferon-gamma signature genes were identified in accordance with previous studies^{24,47}.”

-Online Methods “Pseudo-bulk differential gene expression analysis using scRNA-seq data”

We have further added the following sentences:

(Line 675 - 678)

“We utilized ENCODE and ChEA Consensus TFs from ChIP-X⁵³ to calculate the adjusted p-values for each transcription factor-related pathway of DEGs in CD14 Mono_Activated and CD14 Mono_ISG. DEGs in B cells were analysed using GO Biological Process 2023⁵⁴.”

4. In Figure 5, why did the authors only choose IFN-a for subsequent analysis instead of the other types of interferons? Did the authors measure the concentrations of IFN-b and IFN-g in those 43 patients? If so, do the concentrations of these two types of interferons show correlation with patient symptoms, MPO-ANCA titers and immunotherapy efficacy?

<Our response 2-4>

We appreciate this important suggestion.

i) IFN β

As the reviewer pointed, IFN β can induce ISG. We measured IFN β levels in

eight cases and examined the correlation with ISG scores. As a result, IFN β levels did not correlate with ISG scores (Extended Data Fig. 11b). Therefore, we did not incorporate IFN β as a parameter in the subsequent cohort of N = 43.

ii) IFN γ

Our pathway analysis (Fig. 3a) indicates that ISGs are stimulated by type I interferons, IFN α and IFN β . Consequently, IFN γ , a type II interferon, was initially considered irrelevant and excluded. In response to the reviewer's advice on the importance of verification, we created a correlation graph between the IFN γ (type II) signature score and ISG (type I) score, confirming the lack of relevance of IFN γ in this context. (Extended Data Fig. 11c).

<Revised points>

-Extended Data Figures

Extended Data Fig. 11b was added, which shows the correlations between serum FN- β concentrations and interferon signature gene scores.

Extended Data Fig. 11c was added, which shows the correlations between IFN- γ signature gene score and interferon signature gene scores.

Minor issue:

The authors should provide figures with better resolution. Some genes and labels in several panels such as Fig. 3b are rather blurry to be clearly recognizable.

<Our response 2-5>

We apologize that some parts of the figures were visually difficult to catch. We have addressed this issue by increasing the font size within each figure. Additionally, we have uploaded high-resolution versions of the figures as separate files, in addition to figure-embedded version of the manuscript.

<Revised points>

- Figures

The font size within each figure was increased (Fig. 1, 2, 3, 4, 5, and extended data figures).

REVIEWERS' COMMENTS

Reviewer #1 (Remarks to the Author):

The authors answered all raised points sufficiently

Reviewer #2 (Remarks to the Author):

The authors have adequately addressed my concerns.